# Towards Out-of-Modal Generalization without Instance-level Modal Correspondence

**Zhuo Huang**[1], **Gang Niu**[2], **Bo Han**[3,2], **Masashi Sugiyama**[2,4], **Tongliang Liu**[1,2,*]

[1]Sydney AI Centre, The University of Sydney  [2]RIKEN Center for Advanced Intelligence Project
[3]Hong Kong Baptist University  [4]The University of Tokyo

## ABSTRACT

The world is understood from various *modalities*, such as appearance, sound, and language. Since each modality only partially represents objects in a certain meaning, leveraging additional ones is beneficial in both theory and practice. However, exploiting novel modalities normally requires cross-modal pairs corresponding to the same instance, which is extremely resource-consuming and sometimes even impossible, making knowledge exploration of novel modalities largely restricted. To seek practical multi-modal learning, here we study *Out-of-Modal (OOM) Generalization* as an initial attempt to generalize to an unknown modality without given instance-level modal correspondence. Specifically, we consider Semi-Supervised and Unsupervised scenarios of OOM Generalization, where the first has scarce correspondences and the second has none, and propose *Connect&Explore* (COX) to solve these problems. COX first connects OOM data and known In-Modal (IM) data through a variational information bottleneck framework to extract shared information. Then, COX leverages the shared knowledge to create emergent correspondences, which is theoretically justified from an information-theoretic perspective. As a result, the label information on OOM data emerges along with the correspondences, which helps explore the OOM data with unknown knowledge, thus benefiting generalization results. We carefully evaluate the proposed COX method under various OOM generalization scenarios, verifying its effectiveness and extensibility. The code is available at https://github.com/tmllab/2025_ICLR_COX.

## 1 INTRODUCTION

To understand the world, we use various data *modalities*, such as image (He et al., 2016; 2017; Ren et al., 2015) and text (Devlin et al., 2018; Vaswani et al., 2017). Each modality describes objects through a certain physical perspective, thus contributing to understanding objects. Therefore, *multi-modal learning* (MML) (Alayrac et al., 2022; Ngiam et al., 2011; Radford et al., 2021; Socher et al., 2013) which learns from multiple modality data has been a core research topic in AI. Thanks to the utilization of various modalities, the learning performance has shown to be beneficial on various tasks compared to uni-modal learning (Huang et al., 2021; Lu, 2024; Radford et al., 2021; Sun et al., 2020), such as cross-modal retrieval and generation (Yasunaga et al., 2023; Zhang et al., 2021; Zhen et al., 2019), human-computer interaction (Pantic & Rothkrantz, 2003; Rahman et al., 2022), and robotics (Jiang et al., 2023; Yu et al., 2023).

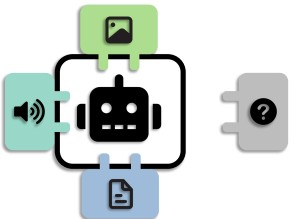

Figure 1: AI is enhanced as more modalities are incorporated, so how can we teach AI to learn from novel modalities based on the ones it already know?

However, existing states of the art are not satisfactory, and emerging modalities need to be leveraged effectively just like the relatively new data modalities of geomagnetic fields (Hashimoto, 1926), sound waves (Harley et al., 2003), and electromagnetic waves (Weinstein, 1988). Therefore, emerging technologies have constantly leveraged new sensors to enhance their performance. For example, Embodied AIs (Savva et al., 2019) already possess abilities like 3D vision and language, but they are still exploring novel skills, such as tactile and bio-sensing. Since it is hard to leverage such uncommon and inexperienced skills in practice, adapting the knowledge from common modalities to

---

*Corresponding to Tongliang Liu (tongliang.liu@sydney.edu.au).

understand the novel ones could be beneficial, as shown in Figure 1. In practice, most existing MML investigations (Radford et al., 2021; Girdhar et al., 2023; Wang et al., 2024d; Zhu et al., 2023) require *instance-level modal correspondence*, *i.e.*, multi-modal data are paired with the same instance, which is often hard to satisfy in real-world scenarios when facing novel modalities (Liang et al., 2023; 2021; Sun et al., 2020; Xia et al., 2024). For a robotic example, some modalities are common and easy to acquire, *e.g.*, vision and language. However, others like tactility need special sensors to resample from the same objects seen or spoken. Unfortunately, the resample could no longer be accessible in practice. As a result, the new modalities usually have incomplete or even no correspondence, which could seriously block the knowledge interaction across modalities and hinder the benefits brought by MML. Hence, a question naturally occurs: *Do we really need instance-level modal correspondence to explore novel modalities?*

This paper studies a practical yet unexplored problem named *Out-of-Modal (OOM) Generalization*. Particularly, given several modalities, *i.e.,* In-Modal (IM) data, the goal is to generalize to an unknown modality without or sometimes only with scarce correspondence. Such a setting implies the real-world utilization of novel modalities: Even though human perception is limited to certain modalities, *e.g.*, touch, sight, sound, and smell, we can still understand unperceivable ones such as magnetism by utilizing inherently-possessed senses, *e.g.*, feel the force when pulling two magnets together; or see the magnetic field by observing the alignment of iron filings around a magnet.

Based on this insight, we utilize IM perceptors that contain prior knowledge to encode known IM data, which can be implemented using existing MML models (Radford et al., 2021; Girdhar et al., 2023; Zhu et al., 2023; Wang et al., 2024d), and an OOM learner which learns novel modalities without any prior knowledge. By analyzing the interactions between latent features, we show theoretically and empirically that the OOM learner can be trained to gradually discover the OOM knowledge, as shown in Figure 2. First, we consider *semi-supervised OOM generalization* where few correspondences are given. Based on the correspondence, we can capture the prior probability distribution and learn mappings that connect OOM data and IM data. Through an information-theoretic perspective, we propose *Connect&Explore* (COX), which encourages the agreement on mappings across modalities, further sharing the cross-modal knowledge and exploring OOM knowledge. Then, we extend COX to an *unsupervised OOM generalization* scenario where there is no instance-level correspondence at all. To tackle such a challenge, we enhance the OOM-IM connections by maximizing cross-modal interaction. First, we select data pairs from cross-modal mappings according to feature similarity. By assuming that the data pairs closing to OOM mappings can be considered as correspondence, we can create emerging correspondence and solve the unsupervised case via the semi-supervised solution. To validate the proposed COX, we carefully design experiments using various multi-modal datasets to validate its effectiveness. Moreover, we provide extensive analyses in various scenarios to understand our method and inspire future research. To sum up, our contributions are three-fold:

- We discover a novel and practical problem named Out-of-Modal Generalization, which aims to explore a novel modality using the knowledge from known modalities.
- We consider two typical situations: Semi-Supervised OOM generalization and Unsupervised OOM generalization, and propose a Connect&Explore framework to tackle both problems from an information-theoretic perspective.
- We conduct extensive experiments to tackle the OOM generalization on various datasets and provide intuitive insights to help inspire future research.

## 2 RELATED WORK

**Modality Generalization** (Liu et al., 2024) generally focuses on leveraging the knowledge from some modalities and generalizing to another one. Existing studies are conducted in different settings and with various tasks. Cross-modal fine-tuning mimics transfer learning by adapting the distribution of IM data to OOM data using the same model. Shen et al. (2023) proposed to conduct distribution alignment to achieve this goal which requires both pre-trained knowledge and labeled target modality data. Based on a similar problem setting, Cai et al. (2024c) designed a gradual modality generation scheme that selects the top-$k$ active feature patches from target modalities, and replaces them with source modalities patches. Such a progressive strategy can align target modal data to ensure generalization. Cross-Modal Generalization uses separate encoders and focus on generalizing to a different modality data from the same instance. Liang et al. (2021) used meta-learning

Table 1: A comparison of different MML problems and their corresponding settings.

| Problem | References | IM Knowledge | OOM Knowledge | Correspondence |
|---------|-----------|--------------|---------------|----------------|
| Cross-Modal Fine-Tuning | Shen et al. (2023); Cai et al. (2024c) | pre-trained & labeled | labeled | ✗ |
| Cross-Modal Generalization | Liang et al. (2021) | pre-trained & labeled | pre-trained | ✔ |
| | Xia et al. (2024) | pre-trained & labeled | pre-trained & labeled | ✔ |
| MML w/o labeled Multi-Modal Data | Liang et al. (2023) | partially labeled | partially labels | ✔ |
| OOM Generalization | Semi-Supervised case (Section 3.3) | pre-trained & labeled | scarcely labeled | A few |
| | Unsupervised case (Section 3.4) | pre-trained & labeled | ✗ | ✗ |

to align OOM data to IM space and generalize to OOM tasks dynamically. Xia et al. (2024) studied a different setting where IM and OOM data are both known during training. Then, a unified representation space is learned to help with the downstream generalization of OOM data. Some other studies consider generalization when all modalities are available, Ma et al. (2019) studied cross-modal generalization without paired data, Wang et al. (2023) applied the information bottleneck to CLIP training, Fang et al. (2024) conducted multi-modal fusion under limited clinical data, and Dong et al. (2023) considered domain generalization with fully-paired multi-modal data. A recent study MML without Labeled Multi-Modal Data (Liang et al., 2023) proposed a different setting where both IM and OOM data have labels, but they are not paired. Instead, additional unlabeled paired multi-modal data is given for learning the interaction between modalities. Moreover, Xue et al. (2022) understood the interactions and applied it to knowledge distillation. Except for cross-modal fine-tuning which follows transfer learning, existing MML works mostly require instance-level correspondence. This work proposes OOM Generalization, where there is no correspondence and the OOM knowledge is barely provided. The comparison of related works is shown in Table 1.

**Modality Binding** aims to learn a joint embedding space across different modalities. Contrastive Language-Image Pre-training CLIP (Radford et al., 2021) is the first work that aligns image with language data. Then, ImageBind (Girdhar et al., 2023) proposed to use vision modalities to bind various modalities into the same representation space. Further, LanguageBind (Zhu et al., 2023) proposed using language as an alternative solution, which binds various modalities similarly. Recently, FreeBind (Wang et al., 2024d) extended the existing unified space into an additional expert space. Specifically, two types of binding were considered, namely space displacement bond and space combination bind. Since modality binding often requires a large amount of data with correspondence, the selected modalities are often quite common. Therefore, the OOM generalization problem can take advantage of the development of modality binding by leveraging the encoders as our IM perceptors to learn novel modalities.

## 3 OOM GENERALIZATION

In this section, we first formalize the OOM generalization setting. Then, we demonstrate the proposed method. Further, we consider a Semi-Supervised case where a few correspondences are available and an Unsupervised scenario where there is no correspondence, showing that the proposed method can successfully tackle both settings and effectively leverage unpaired OOM data.

### 3.1 PROBLEM SETTING

In OOM generalization, we are given a set of known modalities $\{\mathcal{M}_1^{\mathrm{I}}, \ldots, \mathcal{M}_K^{\mathrm{I}}\}$ where $\mathcal{M}_{k \in \{1,\ldots,K\}}^{\mathrm{I}} = \{(x_{k,i}^{\mathrm{I}}, y_{k,i}^{\mathrm{I}})_{i=1}^N \in \mathcal{X} \times \mathcal{Y}\}$ is composed of $N$ number of labeled IM examples with its subscript $i$ denoting the correspondence across different modalities. Moreover, we have an unknown modality $\mathcal{M}^{\mathrm{O}} = \{(x_j^{\mathrm{O}})_{j=1}^M\}$ containing $M$ unlabeled OOM examples. In some cases, it is possible to obtain few correspondences with IM data, then our OOM data could be $\mathcal{M}^{\mathrm{O}} = \{(x_i^{\mathrm{O}}, y_i^{\mathrm{O}})\}_{i=1}^L \cup \{(x_j^{\mathrm{O}})\}_{j=L+1}^M$, where $L \ll M$ and the subscript $i$ traces the corresponding IM data instance and label.

To tackle OOM generalization, we propose a learning framework as shown in Figure 2. Particularly, we use a set of IM perceptors $\{g_1^{\mathrm{I}}, \ldots, g_K^{\mathrm{I}}\}$ to perceive IM data, which can be realized by many existing modality-binding models, such as ImageBind (Girdhar et al., 2023) and LanguageBind (Zhu et al., 2023).

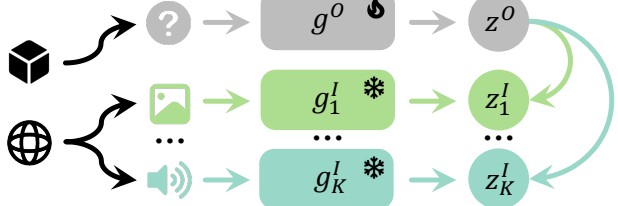

Figure 2: Learning framework of our OOM generalization.

Then, the features of IM data are obtained via $z_k^I = g_k^I(x_k^I)$. Moreover, we use an OOM learner $g^O$ to learn features $z^O$ from OOM data through $z^O = g^O(x^O)$. Our goal is to effectively generalize to OOM data by exploring the relationships between the OOM feature $z^O$ and IM features $\{z_k^I\}_{k=1}^K$. Note that we only focus on the generalization performance of OOM data, the improvement of learning IM data is not the goal of this paper. Therefore, we freeze the parameters of all IM perceptors and only train the OOM learner during experiments. On top of the above models, we further define classifiers $h^O(x^O) := h^O(x^O; g^O)$ and $h_k(x_k^I) := h_k(x_k^I; g_k^I)$ that make predictions.

## 3.2 METHODOLOGY: CONNECT&EXPLORE (COX)

Here we elucidate the proposed method based on the interactive relationship between modalities (Liang et al., 2023; Williams & Beer, 2010). Specifically, the total information of two modalities under a certain task is decomposed into 1) *commonality*[1] which indicates common attributes across modalities, 2) *uniqueness* that is only presented in each modality, and 3) *synergy* denoting the emerging information when modalities are presented together. Note that we do not consider 3) in this paper as our goal is generalizing to OOM data.

To generalize to an unknown modality based on common ones, we aim to extract the commonality that can help partially comprehend OOM data based on IM data. Then, we model the posterior distribution of OOM data by selecting anchor points with minimum uniqueness. To this end, the OOM generalization can be successfully established. The proposed COX method comprises two steps: 1) learning connections by mapping IM data to OOM data to extract commonality, and 2) exploring high uniqueness OOM data by matching their posterior to high-commonality OOM data.

**Connection through Commonality** aims to capture common knowledge across modalities using generative models (Lu, 2024). Here we follow the variational information bottleneck (VIB) framework (Alemi et al., 2016) to achieve this goal. We assume that given IM data $X^I$ and OOM data $X^O$, the latent variable $V$ extracted from $X^I$[2], and label $Y$, the joint distribution is factorized as

$$p(X^I, X^O, V, Y) = p(V, Y|X^O, X^I)p(X^O|X^I)P(X^I), \qquad (1)$$

where we assume $p(V, Y|X^O, X^I) = p(V|X^I)p(Y|X^I)$, corresponding to the Markov chains $V \leftrightarrow X^I \leftrightarrow X^O$ and $X^I \leftrightarrow Y \not\leftrightarrow X^O$. Such an assumption means that $V$ is not related to $X^O$ (Alemi et al., 2016) and the given label $Y$ is not directly connected to $X^O$ under our OOM setting. Intuitively, given an IM datum, *i.e.*, dog image, it is sufficient to infer the label "dog", and the same for inferring from an unknown OOM datum, *i.e.*, dog bark. Thus, in common multi-modal settings, the label prediction using IM information dog image is not further conditioned on OOM knowledge dog bark, because here the OOM knowledge is redundant when IM data is given.

Our goal is to extract valuable knowledge from IM data to leverage OOM data by maximizing the information commonality (Liang et al., 2023; Williams & Beer, 2010):

$$\max I(X^O; X^I; Y) = I(X^O; X^I) - I(X^O; X^I|Y), \qquad (2)$$

where $I(X^O; X^I; Y)$ denotes the mutual information between $X^O$ and $X^I$ regarding the task $Y$, *i.e.*, the label; and $I(X^O; X^I|Y)$ indicates the conditional mutual information irrelevant to $Y$. We start with the first term:

$$I(X^O; X^I) = \int dx^O dx^I p(x^O, x^I) \log \frac{p(x^O, x^I)}{p(x^O)p(x^I)} = \int dx^O dx^I p(x^O, x^I) \log \frac{p(x^O|x^I)}{p(x^O)}, \qquad (3)$$

where $p(x^O|x^I) = \int dv p(x^O, v|x^I) = \int dv p(x^O|v)p(v|x^I)$ can be approximated via a decoder $q(x^O|v)$. Since the Kullback Leibler (KL) divergence is always non-negative, we have $KL[p(X^O|V) \parallel q(X^O|V)] \geq 0 \Rightarrow \int dx^O p(x^O|v) \log p(x^O|v) \geq \int dx^O p(x^O|v) \log q(x^O|v)$, and leveraging Jensen's inequity, we can have

$$I(X^O; X^I) \geq \int dx^O dx^I p(x^O, x^I) \log \frac{\int dv q(x^O|v)p(v|x^I)}{p(x^O)} \qquad (4)$$

$$\geq \int dx^O dx^I dv p(x^O, x^I) \log q(x^O|v)p(v|x^I) + H(X^O), \qquad (5)$$

---

[1] It is originally termed "redundancy" which is negative. However, such property is quite positive for tackling our problem, and hence we rename it "commonality".

[2] Note that the latent variable $V$ here is different from the feature representation $z^I$ and $z^O$.

where the last term is independent of our optimization process. Further, we rewrite $p(x^O, x^I) = \int dv p(x^O, x^I, v) = \int dv p(x^I) p(x^O|x^I) p(v|x^I)$. Then, we have the following lower bound:

$$I(X^O; X^I) \geq \int dx^O dx^I dv p(x^I) p(x^O|x^I) p(v|x^I) \log q(x^O|v) p(v|x^I), \quad (6)$$

which is realized by sampling from the joint data distribution, the latent variable from our encoder $p(v|x^I)$, and the tractable variational approximation $q(x^O|v)$.

Similarly, we can upper-bound the second term $I(X^O; X^I|Y)$ (details shown in Appendix A):

$$I(X^O; X^I|Y) \leq \int dx^O dx^I dy p(x^O, x^I, y) \log p(y|x^I) p(x^O|x^I) p(x^I) - \log h^O(y|x^O), \quad (7)$$

where $h^O(y|x^O)$ is our classifier model for predicting OOM data. To this end, we can lower-bound our objective by combining Eqs. 6 and 7:

$$\begin{aligned}
I(X^O; X^I; Y) \geq &\int dx^O dx^I dv p(x^I) p(x^O|x^I) p(v|x^I) \log q(x^O|v) p(v|x^I) \\
&- \int dx^O dx^I dy p(x^O, x^I, y) \log p(y|x^I) p(x^O|x^I) p(x^I) + \log h^O(y|x^O) = \mathcal{L}_{\text{con}}.
\end{aligned} \quad (8)$$

The above lower bound contains two parts: 1) OOM data reconstruction where we reconstruct $X^O$ using the latent $V$ and 2) OOM data label prediction where we model the label distribution $Y$. In practice, we can approximate $p(x^O, x^I, y)$ using empirical samples from IM and OOM data. Moreover, we use encoder $p(v|x^I)$ without any prior assumptions because we can leverage the feature distribution from the pre-trained IM perceptors. Additionally, a classifier $h(y|x^O)$ is optimized to categorize OOM data based on given labels. Empirically, we can minimize

$$\mathcal{L}_{\text{con}} := \frac{1}{M} \sum_{i=1}^{M} \|x_i^O - q(x_i^O|v_i) p(v_i|x_i^I)\|_2^2 - \log h^O(y_i|x_i^O), \quad (9)$$

where we use the reconstruction error $\| \cdot \|_2^2$[3] to realize the log-likelihood $q(x^O|v) p(v|x^I)$, as similarly done by Kingma & Welling (2013). After building the connections, we can ensure the task-relevant information shared across modalities is learned, which helps partially understand OOM data regarding its commonality. However, note that the second term in Eq. 8 is not fully leveraged which contains $p(y|x^I)$ modeled by the IM perceptors. Take a step further, we can obtain $- \int dx^O dx^I dy p(x^O, x^I, y) \log \frac{p(y|x^I) p(x^O|x^I) p(x^I)}{h^O(y|x^O)}$. Since $p(x^O|x^I) p(x^I)$ is fixed in label prediction, we can derive $-\text{KL}(p(y|x^I) \| h^O(y|x^O))$ which implies that the label information related IM data can be harnessed to explore commonality. Next, we demonstrate how the commonality helps OOM generalization, and provide a solution to explore uniqueness.

**Exploration of Uniqueness** can be achieved via selecting and exploring the OOM data with high uniqueness. To identify these data, we can leverage the agreement and disagreement achieved by the optimal classifiers from various IM data. Our final goal is to optimize via

$$\min_{h^O} \text{KL}(h^O(y|x_d^O) \| h^O(y|x_a^O)), \text{where } x_d^O \in \mathcal{D}, x_a^O \in \mathcal{A}, \quad (10)$$

in which $h_1^*$ and $h_2^*$ denote the optimal classifiers found in two IM data $x_1^I$ and $x_2^I$, respectively, and $x_d^O$ and $x_a^O$ are selected from OOM data with modality disagreement $\mathcal{D} := \{x^O : h_1^*(x^O) \neq h_2^*(x^O)\}$ and agreement $\mathcal{A} := \{x^O : h_1^*(x^O) = h_2^*(x^O)\}$, respectively. Here we use two in-modalities for simplicity, but the conclusion can be extended to multiple modalities. Moreover, the data with agreement is considered anchor points that guide the exploration of those with disagreement. This objective aims to match the posterior of OOM data with uniqueness $h^O(y|x_d^O)$ to the one of anchor points $h^O(y|x_a^O)$. To justify this, we first define modality disagreement:

**Definition 1** (Modality disagreement). Given $X_1, X_2$ and target $Y$, as well as their corresponding optimal classifiers $h_1^*$ and $h_2^*$, their modality disagreement is defined as $\alpha(h_1^*, h_2^*) = \mathbb{E}_{p(x_1, x_2)}[d(h_1^*, h_2^*)]$ where $d : \mathcal{Y} \times \mathcal{Y} \to \mathbb{R}^+$ is a distance function in the label space scoring the disagreement between $h_1^*$ and $h_2^*$.

---

[3]Though training generative models in input space is computationally inefficient, we propose to connect modalities in the feature space in experiments. Therefore, the raw data $x$ is replaced by latent feature $z$.

**Theorem 1.** Given two Bayes' optimal classifiers $h_1^*$ and $h_2^*$ from two in-modalities, under relaxed triangle inequality, inverse Lipschitz condition, and classifier optimality assumptions (Sridharan & Kakade, 2008), the modalities disagreement is upper-bounded by (see details in Appendix B)

$$\alpha(h_1^*, h_2^*) \leq I(X^O, X_2^I, Y|X_1^I) + I(X^O, X_1^I, Y|X_2^I) + 2I(X^O, Y|X_1^I, X_2^I). \quad (11)$$

Finally, based on the decomposition of the task-related mutual information of $X^O$: $I(X^O, Y) = I(X^O, X_2^I, Y|X_1^I) + I(X^O, X_1^I, Y|X_2^I) + I(X^O, Y|X_1^I, X_2^I) + I(X^O, X_1^I, X_2^I, Y)$, as shown in Figure 3, we can achieve

$$\alpha(h_1^*, h_2^*) \leq I(X^O, Y) - I(X^O, X_1^I, X_2^I, Y) + I(X^O, Y|X_1^I, X_2^I), \quad (12)$$

where the first term denotes the overall information, the second term indicates the commonality shared between all modalities, and the third term stands for the uniqueness only preserved in OOM data. Intuitively, when we try to increase the modality disagreement, the commonality is decreased and OOM uniqueness is increased, which successfully justifies our learning objective: In order to explore the uniqueness of OOM data, we can explore the ones with high modality disagreement; conversely, the OOM data with high commonality and low uniqueness is found where agreement is achieved among $h_1^*$ and $h_2^*$. Therefore, we select such data as anchor points that provide informative guidance to help explore uniqueness.

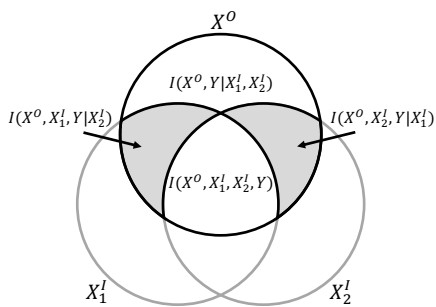

Figure 3: Decomposition of $I(X^O, Y)$.

Next, we consider two realistic scenarios of OOM generalization and demonstrate how the proposed COX method can tackle them.

### 3.3 SEMI-SUPERVISED OOM GENERALIZATION

We start with a semi-supervised case where a few correspondences are available in OOM data, as shown in Figure 4 (a). Based on the VIB framework proposed in Section 3.2, we first leverage the OOM data $\{(x_i^O, y_i^O)\}_{i=1}^L$ corresponding to IM data $\{(x_{k,i}^I, y_{k,i}^O)\}_{i=1}^L, \forall k \in \{1, \ldots, K\}$ to build $K$ connections using additional generative models that can be trained via a point-to-point mapping. As a result, the map-

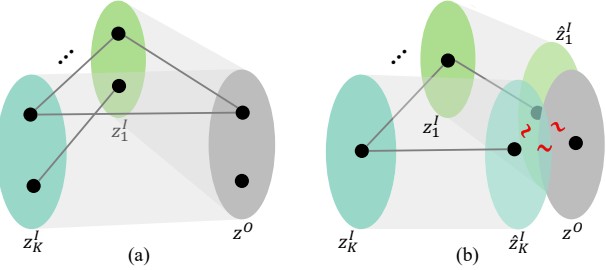

Figure 4: Two scenarios: (a) Semi-Supervised OOM Generalization and (b) Unsupervised OOM Generalizaiton.

pings on the OOM feature space can successfully match the OOM feature distribution, which allows us to directly apply IM data posteriors to select and explore the uniqueness of OOM data. Hence, we formulate our objective as

$$\min_{h^O} \mathcal{L}_{ssl} := \frac{1}{L} \sum_{i=1}^L \mathrm{CE}(h^O(x_i^O), y_i^O) + \frac{1}{L+|\mathcal{D}|} \sum_{x_{d,j} \in \mathcal{D}} \sum_{x_{i=1}^L} \mathrm{KL}(h^O(x_{d,j}^O) \| h^O(x_i^O); h_1^*, h_2^*), \quad (13)$$

where the first term exploits labeled OOM data with correspondence and the second term explores OOM data $\mathcal{D}$ with modality disagreement by minimizing its KL divergence from the label posterior. Through the above objective, we can maximally exploit the uniqueness of OOM data to achieve effective OOM generalization.

### 3.4 UNSUPERVISED OOM GENERALIZATION

As for the unsupervised case, we propose two-phase training: 1) we first conduct a warm-up training to initialize the OOM feature space and the connection, and 2) then, we enhance the connection by creating emergent correspondence and further exploring OOM data.

Specifically, we select anchor points from OOM data by directly applying modality agreement among all Bayes' optimal classifiers from IM data via

$$\mathcal{A}_{\text{sorted}} = \text{SORT}_T(\mathcal{A}, \frac{1}{K}\sum_{k=1}^{K}\max h_k^*(x^{\text{O}})), \text{where } \mathcal{A} = \{\forall x^{\text{O}} \in \mathcal{M}^{\text{O}}: h_1^*(x^{\text{O}}) = \cdots = h_K^*(x^{\text{O}})\}, \quad (14)$$

where the $\text{SORT}_T(\cdot, \cdot)$ is a sort function, which ranks each element $x^{\text{O}}$ in $\mathcal{A}$ based on the value of $\frac{1}{K}\sum_{k=1}^{K}\max h_k^*(x^{\text{O}})$ from large to small. Here, we select anchor points with the top-$T$ largest likelihood averaged over all $K$ IM classifiers. Then, we warm up the OOM learner via minimizing cross-entropy loss $\min \frac{1}{T}\sum_{x^{\text{O}} \in \mathcal{A}_{\text{sorted}}} \text{CE}(h^{\text{O}}(x^{\text{O}}), \arg\max h_k^*(x^{\text{O}}))$. Additionally, we also warm up the connection by leveraging class-wise information. Specifically, we compute the cluster centroids for each modality via $\frac{1}{|\mathcal{C}_y|}\sum_{x_i^{\text{O}} \in \mathcal{C}_y := \{x^{\text{O}}: h^{\text{O}}(x^{\text{O}})=y, y \in \mathcal{Y}\}} z_i^{\text{O}}$ and pair them to each IM centroid correspondingly. To this end, we can build up initial connections by following the VIB framework.

After the warm-up, we aim to further enhance both our connection and OOM exploration by creating emergent correspondence, as shown in Figure 4 (b). To tackle this, we map all IM data into the OOM feature space. If an OOM feature is close to all mappings $v_{k,i}, \forall k = \{1, \ldots, K\}$, then they can form a strong correspondence. Further, we select such OOM data as anchor points, which is further labeled the same as the corresponding IM data. Formally, we optimize OOM learners via

$$\min_{h^{\text{O}}} \mathcal{L}_{\text{uns}} := \frac{1}{|\mathcal{A}|}\sum_{(x_a^{\text{O}},y) \in \mathcal{A}} \text{CE}(h^{\text{O}}(x_a^{\text{O}}), y) + \frac{1}{|\mathcal{A}|+|\mathcal{D}|}\sum_{x_d^{\text{O}} \in \mathcal{D}}\sum_{x_a^{\text{O}} \in \mathcal{A}} \text{KL}(h^{\text{O}}(x_d^{\text{O}}) \| h^{\text{O}}(x_a^{\text{O}}); h_1^*, h_2^*), \quad (15)$$

where $\mathcal{A}$ denotes the updated anchor points which are realized by sorting the Euclidean distance: $\mathcal{A} := \text{SORT}_S(\{(x_j^{\text{O}}, y_i^{\text{I}})\}_{j=1}^M, -\min_{i \in \{1,\ldots,N\}} \frac{1}{K}\sum_{k=1}^{K}\|z_j^{\text{O}} - v_{k,i}\|)$, where the first term computes the cross-entropy loss from the anchor points, and the second term calculates the KL divergence between the OOM data with modality disagreement and the OOM anchor points.

After these two steps, we can effectively tackle the unsupervised OOM generalization. In practice, we connect modalities and select anchor points in the feature space, and hence our application to both two scenarios can be efficient. In the next section, we carefully conduct extensive experiments to justify the effectiveness and extendibility of the proposed COX method under various settings.

## 4 EXPERIMENTS

In our experiments, we first elucidate the experimental details. Then, we provide performance comparisons to various baseline methods on different datasets. Finally, we conduct empirical analyses to provide an intuitive understanding of the proposed method.

### 4.1 IMPLEMENTATION DETAILS

**Datasets.** We consider datasets with at least three modalities: 1) TVL dataset (Fu et al., 2024) contains tactile sensing, RGB image, and class name which can be transformed into language; 2) LLVIP (Jia et al., 2021) dataset has infrared thermal data, RGB image, and annotations for pedestrian detection. We follow Zhu et al. (2023) to crop the pedestrian and background which stand for two classes. Further, we use the OpenAI template (Radford et al., 2021) to create language description; 3) NYU-D dataset (Silberman et al., 2012) contains RGB image, depth data, and class name that can be transformed into language description as well; 4) VGGS dataset (Chen et al., 2020a) includes video data, corresponding sound, and the language description; 5) MSR-VTT (Xu et al., 2016) includes videos and text description, we break down the videos into video frames and the audio data; 6) MOSEI dataset (Zadeh et al., 2018) contains videos from 7 classes of emotions, we extract audio data from the videos and use the emotion type to create language descriptions.

**Models.** We employ two types of IM perceptors, namely ImageBind (Girdhar et al., 2023) and LanguageBind (Zhu et al., 2023) which correspondingly contain 6 and 5 encoders to process different modalities. We select one modality for each experiment as OOM and then choose the rest as IM. For IM data, we use the existing encoders to extract their features. As for OOM data, we conduct preprocessing to ensure its compatibility. Then, we initialize an OOM learner from scratch using ViT-T/16 to learn from the OOM data using the guidance from IM perceptors. Note that for the TVL dataset, there are no existing encoders to process tactile modality. Therefore, when the tactile modality is chosen as IM data, we fine-tune the encoder using contrastive learning on the training

Table 2: Classification performance comparison of different methods across multiple datasets with different OOM modalities.

| Setting | IM Perceptor | Method | TVL | | | LLVIP | | | NYU-D | | | VGGS | | |
|---|---|---|---|---|---|---|---|---|---|---|---|---|---|---|
| | | | RGB | Lan | Tac | RGB | Lan | The | RGB | Dep | Lan | Aud | Vid | Lan |
| Semi-Supervised | ImageBind | Random | 0.4 | 0.3 | 0.2 | 48.2 | 47.3 | 51.0 | 10.2 | 11.3 | 10.2 | 0.3 | 0.3 | 0.3 |
| | | ERM | 23.1 | 19.5 | 22.7 | 54.6 | 53.1 | 54.1 | 45.2 | 44.5 | 38.1 | 9.3 | 10.2 | 8.4 |
| | | EntMin | 24.0 | 21.8 | 23.6 | 56.7 | 57.0 | 55.4 | 48.0 | 46.3 | 39.3 | 10.5 | 13.3 | 8.9 |
| | | COX | **31.2** | **25.3** | **26.5** | **59.2** | **58.3** | **58.3** | **52.3** | **50.7** | **44.2** | **16.8** | **18.4** | **11.7** |
| | | aligned | 79.5 | 29.8 | 35.8 | 65.4 | 61.8 | 63.4 | 61.8 | 54.0 | 52.7 | 27.8 | 29.3 | 19.1 |
| | LanguageBind | Random | 0.4 | 0.3 | 0.2 | 48.2 | 47.3 | 51.0 | 10.2 | 11.3 | 10.2 | 0.3 | 0.3 | 0.3 |
| | | ERM | 23.6 | 20.1 | 22.6 | 56.5 | 54.9 | 58.3 | 44.8 | 44.5 | 39.9 | 9.8 | 13.7 | 9.9 |
| | | EntMin | 25.7 | 23.1 | 25.1 | 59.8 | 60.0 | 62.2 | 49.4 | 47.3 | 42.7 | 11.9 | 14.5 | 12.8 |
| | | COX | **33.5** | **26.3** | **27.3** | **61.2** | **62.3** | **66.4** | **58.8** | **53.5** | **48.4** | **18.3** | **22.1** | **13.4** |
| | | aligned | 81.6 | 31.2 | 38.3 | 74.1 | 73.2 | 87.2 | 68.6 | 65.1 | 57.7 | 38.6 | 32.5 | 20.9 |
| Unsupervised | ImageBind | Random | 0.4 | 0.3 | 0.2 | 48.2 | 47.3 | 51.0 | 10.2 | 11.3 | 10.2 | 0.3 | 0.3 | 0.3 |
| | | SSL | 6.3 | 4.3 | 5.1 | 52.3 | 56.1 | 52.4 | 14.6 | 13.6 | 18.9 | 2.5 | 6.9 | 3.8 |
| | | COX | **18.9** | **15.4** | **17.1** | **54.8** | **57.2** | **53.8** | **21.7** | **22.0** | **19.5** | **9.3** | **10.2** | **10.5** |
| | | aligned | 79.5 | 29.8 | 35.8 | 65.4 | 61.8 | 63.4 | 61.8 | 54.0 | 52.7 | 27.8 | 29.3 | 19.1 |
| | LanguageBind | Random | 0.4 | 0.3 | 0.2 | 48.2 | 47.3 | 51.0 | 10.2 | 11.3 | 10.2 | 0.3 | 0.3 | 0.3 |
| | | SSL | 6.8 | 6.5 | 5.1 | 54.6 | 57.8 | 53.8 | 16.9 | 18.1 | 16.3 | 7.2 | 5.6 | 4.8 |
| | | COX | **19.3** | **19.2** | **18.6** | **55.0** | **56.4** | **55.7** | **24.5** | **23.1** | **20.4** | **10.0** | **11.6** | **10.4** |
| | | aligned | 81.6 | 31.2 | 38.3 | 74.1 | 73.2 | 87.2 | 68.6 | 65.1 | 57.7 | 38.6 | 32.5 | 20.9 |

set. For ImageBind, the tactile encoder is aligned with the image encoder, and for LanguageBind, it is aligned with the language encoder, which is the same as the original training process. For training the connection between modalities, we employ multi-layer perceptrons to realize the variational information bottleneck framework. Moreover, to obtain the optimal classifier from each in-modality, we utilize the extracted features and train a linear layer as classification heads.

**Setup.** We consider two scenarios of OOM generalization: For the semi-supervised case, we sample $10\%$ of the training data as labeled data with each class having a balanced number of labels. For the unsupervised case, we have no labels at all. For selecting the number of anchor points, we choose the same number of examples for the warm-up and training phases, which is $10\%$ of the total training set. To train the OOM learner, we use the Adam optimizer with an initial learning rate of $1e-3$ with weight decay $1e-5$, and train the model for 50 epochs.

**Baseline methods.** Since there is no existing baseline method to compare with under our setting, we implement four methods for comparison, namely: Random where the model is randomly initialized, ERM where only labeled data is used to minimize the empirical risk, EntMin (Grandvalet & Bengio, 2004) which minimize the entropy of unlabeled data meanwhile conduct ERM, SSL which conducts self-supervised learning using Gaussian noise perturbation on the input, and MoCo He et al. (2020) which updates model parameters with ensembling and meanwhile conducts contrastive learning. Note that we use MoCo for comparison for the retrieval task in Table 3 because it is not for classification, and it is combined with EntMin in the semi-supervised case. Moreover, we use a pretrained encoder as an upper-limit baseline "aligned". Next, we carefully compare the performance of our COX to these baseline methods.

## 4.2 Performance Comparison

For performance comparisons, we conduct classification and cross-modal retrieval to validate the proposed COX. There are seven modalities are considered, namely RGB image, language, tactile, thermal, depth, audio, and video which are simplified as RGB, Lan, Tac, The, Dep, Aud, and Vid, respectively. For each column, we choose one modality as OOM data, the rest modalities are selected IM data. For the retrieval task, we report the recall rate in both top 1 (R@1) and top 5 (R@5). The results are shown in Tables 2 and 3. We can see that the proposed COX clearly shows the best performance in both scenarios. Specifically, COX can achieve more than $5\%$ performance improvement for most of the OOM setting, which justifies that leveraging the knowledge from IM perceptors can indeed help OOM generalization compared to using OOM data alone. Moreover, even though the performance is relatively limited compared to the fully pre-trained baseline under the unsupervised case, considering it is an extremely challenging setting, we can still largely improve the performance for over $10\%$ compared to the Random baseline, which demonstrates that the unsupervised OOM generalization is indeed learnable further leads to a novel research direction for improving the gen-

Table 3: Cross-modal retrieval performance comparison of different methods across multiple datasets with different OOM modalities.

| Setting | IM Perceptor | Method | MSR-VTT Aud R@1 | Aud R@5 | Lan R@1 | Lan R@5 | Vid R@1 | Vid R@5 | MOSEI Aud R@1 | Aud R@5 | Lan R@1 | Lan R@5 | Vid R@1 | Vid R@5 |
|---|---|---|---|---|---|---|---|---|---|---|---|---|---|---|
| Semi-Supervised | ImageBind | Random | 5.4 | 25.1 | 5.0 | 25.4 | 5.4 | 24.2 | 14.3 | 42.5 | 14.4 | 42.8 | 14.1 | 42.1 |
| | | ERM | 15.6 | 30.3 | 16.1 | 35.2 | 18.5 | 38.2 | 28.0 | 45.3 | 29.3 | 47.1 | 33.4 | 48.2 |
| | | EntMin | 18.5 | 32.4 | 19.2 | 38.5 | 21.0 | 39.4 | 29.6 | 46.7 | 32.0 | 48.7 | 35.4 | 50.5 |
| | | MoCo | 20.5 | 33.9 | 21.1 | 38.9 | 23.4 | 43.5 | 30.1 | 47.3 | 32.7 | 50.1 | 36.2 | 51.0 |
| | | COX | **23.3** | **35.8** | **23.4** | **39.1** | **26.5** | **48.8** | **32.4** | **48.0** | **33.8** | **50.4** | **38.8** | **53.7** |
| | | Aligned | 35.5 | 51.5 | 32.3 | 52.4 | 36.8 | 61.8 | 42.9 | 66.4 | 48.2 | 69.4 | 50.5 | 71.6 |
| | LanguageBind | Random | 5.2 | 24.3 | 5.4 | 25.1 | 5.0 | 25.6 | 13.5 | 43.1 | 14.2 | 42.7 | 14.6 | 41.9 |
| | | ERM | 16.3 | 31.1 | 16.5 | 36.2 | 18.7 | 37.9 | 27.3 | 45.5 | 28.4 | 47.6 | 33.4 | 49.3 |
| | | EntMin | 19.6 | 33.4 | 19.8 | 38.6 | 22.4 | 37.9 | 30.2 | 45.5 | 33.5 | 49.0 | 36.0 | 49.7 |
| | | MoCo | 21.1 | 34.8 | 20.9 | 39.2 | 24.5 | 38.6 | 31.1 | 46.7 | 34.5 | **50.5** | 37.0 | 51.7 |
| | | COX | **25.2** | **36.0** | **24.1** | **40.0** | **28.7** | **49.5** | **34.6** | **49.8** | **34.6** | 50.2 | **39.2** | **55.4** |
| | | Aligned | 42.0 | 53.6 | 38.8 | 58.6 | 44.8 | 70.0 | 44.6 | 68.9 | 49.5 | 67.4 | 51.1 | 68.3 |
| Unsupervised | ImageBind | Random | 5.4 | 25.1 | 5.0 | 25.4 | 5.4 | 24.2 | 14.3 | 42.5 | 14.4 | 42.8 | 14.1 | 42.1 |
| | | SSL | 8.9 | 28.4 | 9.3 | 28.1 | 10.1 | 29.5 | 17.4 | 48.8 | 16.2 | 45.2 | 16.0 | 45.0 |
| | | MoCo | 9.2 | 28.9 | 9.5 | 28.4 | 10.6 | 30.0 | 17.8 | 50.3 | 16.6 | 45.8 | 17.1 | 44.4 |
| | | COX | **13.5** | **30.4** | **16.5** | **32.4** | **15.2** | **34.8** | **20.8** | **53.7** | **18.7** | **46.7** | **18.2** | **48.9** |
| | | Aligned | 35.5 | 51.5 | 32.3 | 52.4 | 36.8 | 61.8 | 42.9 | 66.4 | 48.2 | 69.4 | 50.5 | 71.6 |
| | LanguageBind | Random | 5.2 | 24.3 | 5.4 | 25.1 | 5.0 | 25.6 | 13.5 | 43.1 | 14.2 | 42.7 | 14.6 | 41.9 |
| | | SSL | 9.2 | 28.9 | 11.0 | 28.8 | 10.3 | 28.7 | 18.0 | 48.9 | 18.4 | 45.0 | 17.8 | 45.6 |
| | | MoCo | 9.6 | 29.4 | 11.1 | 28.5 | 11.0 | 29.3 | 18.8 | 50.7 | 18.5 | 45.2 | 18.0 | 45.5 |
| | | COX | **14.8** | **31.1** | **18.4** | **34.4** | **15.4** | **35.0** | **23.1** | **52.8** | **19.4** | **47.2** | **20.4** | **49.9** |
| | | Aligned | 42.0 | 53.6 | 38.8 | 58.6 | 44.8 | 70.0 | 44.6 | 68.9 | 49.5 | 67.4 | 51.1 | 68.3 |

eralization performance. Additionally, note that the performance of COX is affected by the quality of IM perceptors, as using LanguageBind shows relatively higher performance compared to using ImageBind. Thus, it would be potentially helpful to leverage sophisticated IM perceptors to benefit the generalization performance.

## 4.3 EMPIRICAL ANALYSIS

To provide an intuitive justification for the proposed method, here we conduct empirical analyses using the MSR-VTT dataset on various OOM scenarios and modalities.

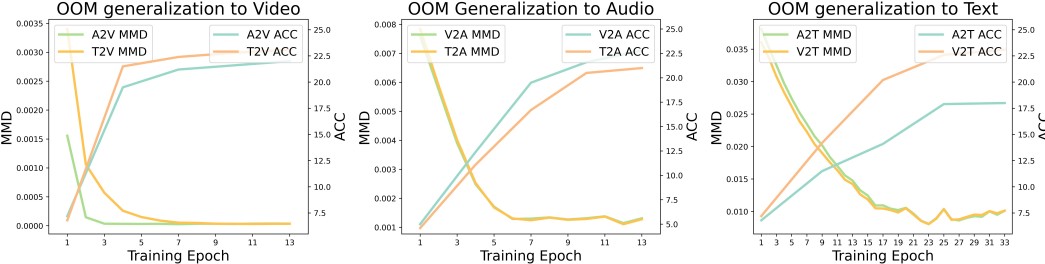

Figure 5: Connection effect on maximum mean discrepancy and accuracy across modalities.

**Connection mitigates modality gap.** To understand the performance of our VIB-based connection learning, here we show its effect on generalization out-of-modal. Specifically, during connection training, we compute the maximum mean discrepancy (MMD) between the mapping of each IM data and the OOM data. Meanwhile, we evenly select 6 points during the training and extract the IM mappings which are used to learn a classification head as the optimal classifier. Based on our theoretical result, we apply the classifiers to OOM data and compute their accuracies, as shown in Figure 5. We can see that as training goes on, the MMD between each IM mapping and OOM data is decreasing and the corresponding accuracy is increasing, which shows that: 1) our connection can indeed close the modality gap between their features and 2) as the mappings of IM data getting close to OOM data, the optimal classifier shows better classification results on OOM data, which benefits the knowledge transfer from known modalities to unknown ones.

**Modality disagreement identifies uncertainty.** To understand the effect of modality disagreement, we conduct an analysis under the semi-supervised scenario by training the OOM learner to use only labeled data for 10 epochs. Then, we leverage the modality disagreement criteria to separate OOM data into those with disagreement and agreement and show their prediction accuracies in

Figure 6 (a). We can see that the accuracy for OOM data with disagreement is significantly lower than those with agreement, meaning that the prediction uncertainty, *i.e.*, data with low accuracy, is effectively identified by the proposed modality disagreement.

**Modality agreement alleviates uncertainty.** Further, we conduct training by following the procedure proposed in Section 3.3 and again show the accuracies of OOM data with disagreement and agreement in Figure 6 (b). We can see that the performance gap between the two types of data is largely mitigated, which justifies the methodology of exploring OOM data using the guidance of modality agreement. As a result, we can achieve almost comparable performance on both types of data, benefiting the overall generalization performance.

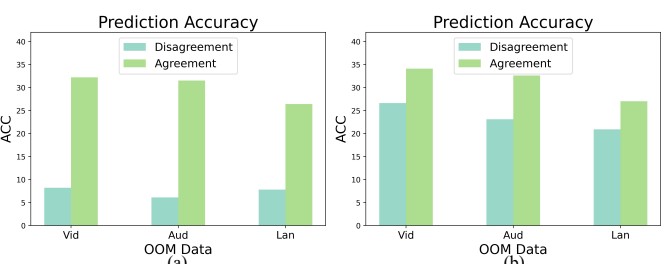

Figure 6: Prediction accuracy of OOM data with modality disagreement and modalities agreement, respectively. (a) Before exploration. (b) After exploration.

**Ablation study.** Additionally, we conduct an ablation study to justify the effect of our methodology. Specifically, we consider three ablations: 1) "w/o connection" where we remove the connection and directly apply the modality disagreement criteria on the original features of IM data and OOM data, 2) "w/o exploration" where we only leverage the OOM data with agreement for training, 3) For unsupervised scenario, we consider "w/o warm-up" where we do not conduct the warm-up phase and directly

Table 4: Ablation study on various settings.

| Setting | Ablation | MSR-VTT R@1 | | |
|---------|----------|------|------|------|
| | | Aud | Lan | Vid |
| Semi | w/o connection | 8.7 | 7.9 | 10.3 |
| | w/o exploration | 16.4 | 16.5 | 18.8 |
| | COX | **25.2** | **24.1** | **40.0** |
| Unsup. | w/o warm-up | 7.4 | 11.5 | 10.5 |
| | COX | **14.8** | **18.4** | **15.4** |

training the model. The results in Table 4 show that all modules are essential for achieving effective OOM generalization. Specifically, the connection is vital for the knowledge transduction of IM data to OOM data, without which the generalization performance is largely degraded. The conclusion is consistent with the connection analysis where directly applying optimal classifiers across modalities leads to poor accuracy. Moreover, removing exploration also hinders the performance because the uniqueness of OOM data is largely ignored. Additionally, we find that the warm-up phase is essential for the unsupervised case. As initialized models have no classification capability, we need pre-training to form basic feature clusters that are consistent with IM data, further enabling effective OOM generalization.

**Discussion on computational efficiency.** Note that we conduct the feature connection mostly on the feature space, the computational cost of training VIB framework work is quite acceptable. The main cost is training the OOM learner which is the basic training with cross-entropy loss optimization and can be implemented on a single NVIDIA 3090/4090 GPU.

## 5 CONCLUSION AND LIMITATION

In this paper, we study a novel and promising research direction dubbed Out-of-Modal (OOM) Generalization which aims to leverage knowledge from existing modalities to generalize to an unknown modality without instance-level correspondence. We consider two scenarios where there are a few correspondences and there is no correspondence, *i.e.*, semi-supervised and unsupervised cases, respectively. To tackle these problems, we propose a Connect&Explore (COX) method which first learns connections across modalities to extract common knowledge and then explores the unique knowledge of OOM data based on modality disagreement. Extensive experiments are conducted to justify the proposed method and intuitive insights are provided to inspire future studies. However, our research is limited to several aspects which we hope to address in the future. First, although challenging as it is, the performance is relatively limited compared to fully-aligned models, which requires more investigations to enhance generalization. Second, our OOM generalization is mostly conducted within the modalities from the same dataset. In the future, we hope to discover scenarios where the OOM data is from a different dataset with a large modality gap.

## 6 ACKNOWLEDGEMENTS

Bo Han was supported by RGC Young Collaborative Research Grant No. C2005-24Y and RIKEN Collaborative Research Fund. Masashi Sugiyama was supported by JST ASPIRE Grant Number JPMJAP2405. Tongliang Liu was partially supported by the following Australian Research Council projects: FT220100318, DP220102121, LP220100527, LP220200949, and IC190100031, he was also supported by JST ASPIRE Grant Number JPMJAP2405.

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

In the appendix, we first provide a detailed derivation of our VIB framework. Then, we carefully prove the modality disagreement in Theorem 1. Further, we provide additional experiments to evaluate our method. Finally, we give a discussion on the relationship between existing studies and provide a prospective outlook for future research.

## A LOWER BOUND OF OUR VIB FRAMEWORK

Recall that we have the following factorization:

$$p(X^{\mathrm{I}}, X^{\mathrm{O}}, V, Y) = p(V, Y | X^{\mathrm{O}}, X^{\mathrm{I}}) p(X^{\mathrm{O}} | X^{\mathrm{I}}) P(X^{\mathrm{I}}), \tag{16}$$

with Markov chains $V \leftrightarrow X^{\mathrm{I}} \leftrightarrow X^{\mathrm{O}}$ and $X^{\mathrm{I}} \leftrightarrow Y \not\leftrightarrow X^{\mathrm{O}}$. Our goal is to maximize the information redundancy (Liang et al., 2023; Williams & Beer, 2010):

$$\max I(X^{\mathrm{O}}; X^{\mathrm{I}}; Y) = I(X^{\mathrm{O}}; X^{\mathrm{I}}) - I(X^{\mathrm{O}}; X^{\mathrm{I}} | Y), \tag{17}$$

where the first term is lower-bounded by:

$$I(X^{\mathrm{O}}; X^{\mathrm{I}}) \geq \int dx^{\mathrm{O}} dx^{\mathrm{I}} dv p(x^{\mathrm{I}}) p(x^{\mathrm{O}} | x^{\mathrm{I}}) p(v | x^{\mathrm{I}}) \log q(x^{\mathrm{O}} | v) p(v | x^{\mathrm{I}}), \tag{18}$$

Then, we consider the second term $I(X^{\mathrm{O}}; X^{\mathrm{I}} | Y)$:

$$I(X^{\mathrm{O}}; X^{\mathrm{I}} | Y) = \int dx^{\mathrm{O}} dx^{\mathrm{I}} dy p(x^{\mathrm{O}}, x^{\mathrm{I}}, y) \log \frac{p(x^{\mathrm{O}}, x^{\mathrm{I}} | y)}{p(x^{\mathrm{O}} | y) p(x^{\mathrm{I}} | y)} \tag{19}$$

$$= \int dx^{\mathrm{O}} dx^{\mathrm{I}} dy p(x^{\mathrm{O}}, x^{\mathrm{I}}, y) \log \frac{p(x^{\mathrm{O}}, x^{\mathrm{I}}, y)}{p(y | x^{\mathrm{O}})} - H(Y) + H(Y | X^{\mathrm{I}}) + H(X^{\mathrm{O}}) + H(X^{\mathrm{I}}). \tag{20}$$

Note that we use the factorization $p(x^{\mathrm{O}}, x^{\mathrm{I}}, y) = p(y | x^{\mathrm{I}}) p(x^{\mathrm{O}} | x^{\mathrm{I}}) p(x^{\mathrm{I}})$, and further ignore the entropy terms[4], then we have:

$$I(X^{\mathrm{O}}; X^{\mathrm{I}} | Y) \leq \int dx^{\mathrm{O}} dx^{\mathrm{I}} dy p(y | x^{\mathrm{I}}) p(x^{\mathrm{O}} | x^{\mathrm{I}}) p(x^{\mathrm{I}}) \log p(y | x^{\mathrm{I}}) p(x^{\mathrm{O}} | x^{\mathrm{I}}) p(x^{\mathrm{I}}) - \log h(y | x^{\mathrm{O}}), \tag{21}$$

which is based on the positivity of KL divergence between our classifier $h(y | x^{\mathrm{O}})$ and $p(y | x^{\mathrm{O}})$.

To this end, we can lower-bound our objective by combining Eqs. 18 and 21:

$$I(X^{\mathrm{O}}; X^{\mathrm{I}}; Y) \geq \int dx^{\mathrm{O}} dx^{\mathrm{I}} dv p(x^{\mathrm{I}}) p(x^{\mathrm{O}} | x^{\mathrm{I}}) p(v | x^{\mathrm{I}}) \log q(x^{\mathrm{O}} | v) p(v | x^{\mathrm{I}}) \tag{22}$$

$$- \int dx^{\mathrm{O}} dx^{\mathrm{I}} dy p(y | x^{\mathrm{I}}) p(x^{\mathrm{O}} | x^{\mathrm{I}}) p(x^{\mathrm{I}}) \log p(y | x^{\mathrm{I}}) p(x^{\mathrm{O}} | x^{\mathrm{I}}) p(x^{\mathrm{I}}) + \log h(y | x^{\mathrm{O}}) = \mathcal{L}_{con}. \tag{23}$$

## B PROOF OF THEOREM 1

*Proof.*

**Assumption 1** (Relaxed triangle inequality). For the distance function $d : \mathcal{Y} \times \mathcal{Y} \rightarrow \mathbb{R}^{+}$, there exists $c_d \geq 1$ such that $\forall \hat{y}_1, \hat{y}_2, \hat{y}_3 \in \hat{\mathcal{Y}} d(\hat{y}_1, \hat{y}_2) \leq c_d(d(\hat{y}_1, \hat{y}_3) + d(\hat{y}_2, \hat{y}_3))$.

**Assumption 2** (Inverse Lipschitz condition). For the function $d$, it holds that $\forall h$, $\mathbb{E}[d(h(x_1, x_2), h^*(x_1, x_2))] \leq |\mathcal{L}(h) - \mathcal{L}(h^*)|$, where $h^*$ is the Bayes optimal classifier on both $x_1$ and $x_2$; and $\mathbb{E}[d(h(x), h^*(x))] \leq |\mathcal{L}(h) - \mathcal{L}(h^*)|$, where $h^*$ is the Bayes optimal classifier on $x$.

**Assumption 3** (Classifier optimality). For any classifiers $h$ in comparison to the Bayes' optimal classifier $h^*$, there exists constants $\epsilon > 0$ such that $|\mathcal{L}(h) - \mathcal{L}(h^*)|^2 \leq \epsilon$.

---

[4]We focus on the optimization of $p(Y | X^{\mathrm{O}})$, and $p(Y | X^{\mathrm{I}})$ is given and frozen in our setting.

To bridge $h_1^*$ and $h_2^*$, we use $h_{1,2}^*$ and $h^*$ to denote the Bayes' optimal classifier on both IM data and all data, respectively. Then, we capture the relationship between the uniqueness of OOM data given both IM data and the difference in their Bayes' optimal prediction errors:

$$|\mathcal{L}(h_{1,2}^*) - \mathcal{L}(h^*)|^2 = |\mathbb{E}_X \mathbb{E}_{Y|X_1^I, X_2^I, X} \circ \ell(h^*(x_1^I, x_2^I, x^O), y) - \mathbb{E}_{X_1^I, X_2^I} \mathbb{E}_{Y|X_1^I, X_2^I} \ell(h_1^*(x_1^I, X_2^I), y)|^2 \tag{24}$$

$$\leq |\mathbb{E}_{Y|X_1^I, X_2^I, X} \circ \ell(h^*(x_1^I, x_2^I, x^O), y) - \mathbb{E}_{Y|X_1^I, X_2^I} \ell(h_1^*(x_1^I, X_2^I), y)|^2 \tag{25}$$

$$\leq \mathrm{KL}(p(y|x_1^I, x_2^I, x^O) \parallel p(y|x_1^I, x_2^I)) \tag{26}$$

$$\leq \mathbb{E}_X \mathrm{KL}(p(y|x_1^I, x_2^I, x^O) \parallel p(y|x_1^I, x_2^I)) \tag{27}$$

$$= I(X^O, Y | X_1^I, X_2^I). \tag{28}$$

Then, we first capture the redundancy between one IM data and OOM data given another IM data:

$$|\mathcal{L}(h_1^*) - \mathcal{L}(h^*)|^2 = |\mathbb{E}_X \mathbb{E}_{Y|X_1^I, X_2^I, X} \circ \ell(h^*(x_1^I, x_2^I, x^O), y) - \mathbb{E}_{X_1^I} \mathbb{E}_{Y|X_1^I} \ell(h_1^*(x_1^I), y)|^2 \tag{29}$$

$$\leq |\mathbb{E}_{Y|X_1^I, X_2^I, X} \circ \ell(h^*(x_1^I, x_2^I, x^O), y) - \mathbb{E}_{Y|X_1^I} \ell(h_1^*(x_1^I), y)|^2 \tag{30}$$

$$\leq \mathrm{KL}(p(y|x_1^I, x_2^I, x^O) \parallel p(y|x_1^I)) \tag{31}$$

$$\leq \mathbb{E}_X \mathrm{KL}(p(y|x_1^I, x_2^I, x^O) \parallel p(y|x_1^I)) \tag{32}$$

$$= I(X^O, X_2^I, Y | X_1^I). \tag{33}$$

Further leveraging triangle inequality through the Bayes' optimal classifier $h^*$ and the inverse Lipschitz condition, we have:

$$\mathbb{E}_{p(x_1^I, x_2^I, x^O)}[d(h_1^*, h_{1,2}^*)] \leq \mathbb{E}_{p(x_1^I, x_2^I, x^O)}[d(h_1^*, h^*)] + \mathbb{E}_{p(x_1^I, x_2^I, x^O)}[d(h^*, h_{1,2}^*)] \tag{34}$$

$$\leq |\mathcal{L}(h_1^*) - \mathcal{L}(h^*)|^2 + |\mathcal{L}(h^*) - \mathcal{L}(h_{1,2}^*)|^2 \tag{35}$$

$$\leq I(X^O, X_2^I, Y | X_1^I) + I(X^O, Y | X_1^I, X_2^I). \tag{36}$$

Symmetrically, we can have $|\mathcal{L}(h_2^*) - \mathcal{L}(h^*)|^2 \leq I(X^O, X_1^I, Y | X_2^I)$ and further obtain $\mathbb{E}_{p(x_2^I, x_2^I, x^O)}[d(h_2^*, h_{1,2}^*)] \leq I(X^O, X_1^I, Y | X_2^I) + I(X^O, Y | X_1^I, X_2^I)$. Then combining with Eq. 36:

$$\mathbb{E}_{p(x_1^I, x_2^I)}[d(h_1^*, h_2^*)] \leq I(X^O, X_2^I, Y | X_1^I) + I(X^O, X_1^I, Y | X_2^I) + 2I(X^O, Y | X_1^I, X_2^I) \tag{37}$$

Finally, based on the decomposition of the task-related mutual information of $X^O$: $I(X^O, Y) = I(X^O, X_2^I, Y | X_1^I) + I(X^O, X_1^I, Y | X_2^I) + I(X^O, Y | X_1^I, X_2^I) + I(X^O, X_1^I, X_2^I, Y)$, as shown in Figure 3, we can achieve:

$$\alpha(h_1^*, h_2^*) := \mathbb{E}_{p(x_1^I, x_2^I)}[d(h_1^*, h_2^*)] \leq I(X^O, Y) - I(X^O, X_1^I, X_2^I, Y) + I(X^O, Y | X_1^I, X_2^I), \tag{38}$$

$$\square$$

## C  ADDITIONAL EXPERIMENTS

We conduct additional experiments to further justify the proposed COX. First, we study the performance benefits brought by COX under various correspondence rates in OOM data. Specifically, we choose MSR-VTT and NYU-D datasets and use Vid and Dep as OOM modalities, respectively, and show the result in Figure 7. First of all, we observe that COX brings more benefits when correspondence is more scarce. This is because sufficient correspondence can maximally uncover the knowledge of OOM data. As correspondence gets less, the knowledge that can be explored from correspondence decreases. However, COX leverages the knowledge from IM data which brings more benefits even with less correspondence. Thus, the increased benefits of COX under low-correspondence scenarios demonstrate its effectiveness in tackling OOM generalization without correspondence.

Moreover, we testify how varied performance levels of IM perceptors could affect the OOM performance. To achieve this, we change the number of IM data in each dataset as $10\%$, $40\%$, $70\%$, and

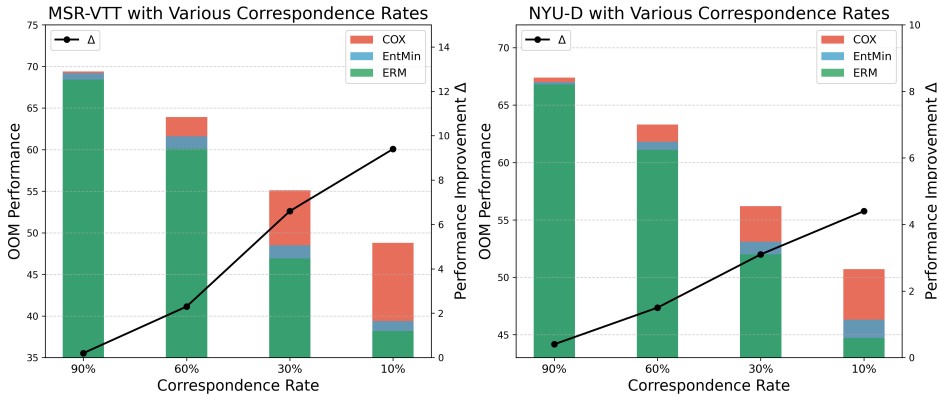

Figure 7: Performance benefits brought by COX under various correspondence rate in OOM data.

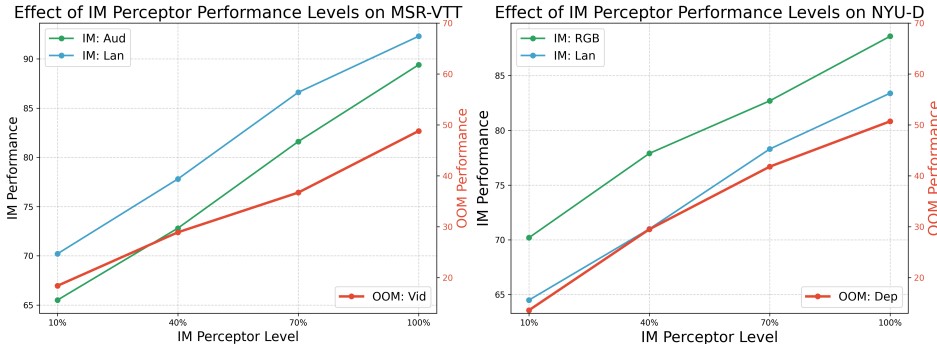

Figure 8: Effect of Varying IM Perceptor Performance Level.

| MSR-VTT | Vision | NYU-D | Language |
|---------|--------|-------|----------|
| FreeMatch | 45.2 | MixText | 21.2 |
| COX | **52.3** | COX | **23.4** |

Table 5: Comparison with competitive uni-modal methods from Vision and Language.

| Setting | Method | MSR-VTT | NYU-D |
|---------|--------|---------|-------|
| Unsup | MoCo | 30.0 | 15.7 |
| | MoCo+COX | **35.4** | **23.8** |

Table 6: Combining COX with MoCo for knowledge extraction from OOM to IM data.

100%, and test the OOM performance of COX, as shown in Figure 8. We can see that the OOM performance is significantly affected by the accuracy level of IM perceptors. When the performance of IM perceptors improves, the OOM performance of COX is also enhanced. Therefore, improving the performance of IM perceptors is vital for OOM generalization using COX.

Further, to understand the contribution of COX on uni-modal study, we conduct comparison and combination with uni-modal methods. First, we consider two uni-modalities vision and language from MSR-VTT and NYU-D datasets, respectively. By comparing to FreeMatch (Wang et al., 2022) and MixText (Chen et al., 2020b), two competitive semi-supervised learning methods that correspondingly deal with vision and language data, we show the performance of COX in Table 5. Even though the two baselines were effective under their original setting, their performance is still limited when applied to challenging multi-modal datasets with scarce knowledge. As we can see, COX still shows very effective performance compared to them, again justifying the benefits from COX by leveraging IM data.

Then, we consider combining COX with the well-known unsupervised method MoCo (He et al., 2020) and show the performance benefits brought by COX for enhancing unknown modality. We show the result in Table 6. Delightfully, we observe significant performance improvement on both

Vid from MSR-VTT and RGB from the NYU-D dataset. This is because COX unleashes the potential label information from IM data to enhance label prediction of OOM data, i.e., Vid and RGB here. Such a finding implies that COX can extract knowledge from other modalities to enhance new ones, which is the main goal of this study.

## D   RETROSPECTIVE AND PROSPECTIVE DISCUSSION

OOM generalization without instance-level correspondence presents challenges that could be related to existing studies but differentiates from them. Here we identify the relationship between OOM generalization and existing fields and discuss how we can be inspired to solve OOM generalization in future studies.

- *Out-of-Distribution (OOD) Generalization* (Hendrycks et al., 2021; Chen et al., 2023c; Huang et al., 2023b; 2024; Wang et al., 2025) where the goal is generalizing to unknown data distributions given several existing ones. The difference between OOM and OOD generalization lies in the data gap: the former faces modality gap while the latter deals with distribution gap. Intuitively, the modality gap demonstrates the change of learning space, thus it is more challenging than the distribution gap where data is still sampled within the same learning space, *i.e.*, same dimension and formats. To solve OOM generalization, a comprehensive understanding of how modalities are generated is required. For example, the surface of an object affects the reflection of light, thus deciding its visual information. Meanwhile, the texture of the surface also influences its tactile sensing. Therefore, we hypothesize that some modalities are commonly influenced by a hidden factor (Glymour et al., 2016; Li et al., 2024b; 2025; Li & Liu, 2025; Lin et al., 2023c; 2025; Zheng et al., 2024). By identifying such a factor, the shared information across modalities could be discovered, further benefiting OOM generalization.

- *Trustworthy Machine Learning* aims to develop reliable, robust, explainable models under realistic scenarios, which considers various types of problems, such as Semi-Supervised Learning (SSL) (Berthelot et al., 2019; Sohn et al., 2020; Huang et al., 2023a; He et al., 2025; Li et al., 2023; 2024a; Wang et al., 2024b), Learning with Noisy Labels (LNL) (Xia et al., 2019; 2020; 2022; Yao et al., 2021; 2020; Yuan et al., 2023; 2024; Wang et al., 2024a; Wu et al., 2024b;a), and Robustness (Rice et al., 2020; Lin et al., 2023a;b; 2024; Hong et al., 2024). Compared to the existing studies, OOM generalization without instance-level correspondence requires linking between data across modalities. During this process, unpaired data and noisy correspondence could occur, further hindering the generalization performance. To solve this problem, progressively generating pseudo correspondence and denoising the noisy ones as done by SSL and LNL might be helpful to complement the correspondence. Therefore, future studies for pairing multi-modal data through trustworthy machine learning techniques could be a promising direction.

- *Foundation Models* (Chen et al., 2023b;a; Radford et al., 2021; Touvron et al., 2023; Bommasani et al., 2021; Liu et al., 2023; Cai et al., 2024b;a) are pre-trained on large-scale data to possess powerful zero-shot generalization abilities (Wang et al., 2024c; Hong et al., 2022; Chen et al., 2022b;a; Tu et al., 2023; 2024a;b; Zheng et al., 2022) have been widely used in practice nowadays. In OOM generalization, we rely on existing foundation models to encode modalities into a shared latent space. Therefore, the performance of our study is highly related to the capabilities of the foundation models. Common strategies either fuse all modalities into a uni-modal model or separately encode and align. Because of the cost of multi-modal correspondence, the former multi-modal fusion strategy might be impractical for large-scale applications. Therefore, alignment between pre-trained encoders could be a promising direction. To effectively organize multiple foundation models to achieve OOM generalization, more comprehensive studies on addressing the modality gap, modality imbalance, and alignment strategy are needed in future research.

