# OpenReview forum: "Towards Out-of-Modal Generalization without Instance-level Modal Correspondence"
_ICLR.cc/2025/Conference — ICLR 2025 Poster_

### Official Review · Reviewer_KNt4 · 2024-10-21

**Soundness:** 3
**Presentation:** 3
**Contribution:** 3
**Rating:** 6
**Confidence:** 4

**Summary:**

This paper propose a novel out-of-modal generalization problem that focuses on generalizing learned knowledge to unknown modalities. In addition, it considers both semi-supervised and unsupervised scenarios. The proposed connect & explore (COX) scheme tries to explore the shared and unique between modalities via variational information bottleneck. The proposed method can build connections between known and unknown modalities, further generating correspondences to explore more knowledge. The expleriments verify the effectiveness of COX.

**Strengths:**

1. This paper proposes a novel OOM problem, which is valuable because of the importance on generalizing exsting knowledge to unknown modalities without paired instances.
2. The proposed COX method sounds reasonable and explores the connections between different modalities without priori knowledge. The motivation and method are clear to me. The results are convincing to me.

**Weaknesses:**

1. I acknowledge the novelty and value of the proposed OOM scenario. However, since the method lacks paired samples, it only shows limited effectiveness in learning shared knowledge, making it significantly inefficient in exploring unknown modalities. This is evident in the experimental results, which show a large gap compared to "aligned". Although the compared methods contain ERM and SSL, I question whether the proposed approach is better than SOTA unsupervised uni-modal solutions. Only surpassing or enhancing existing uni-modal methods with the proposed approach can make it really valuable.
2. As I said in 1, the comparison with existing uni-modal methods is necessary.

**Questions:**

see in Weaknesses

---

> ### Author Response · Authors · 2024-11-21
> **Rebuttal**
>
> We deeply thank reviewer KNt4 for your acknowledgment of our novelty and contribution. Here we carefully address each concern and hope that you will find our reply satisfactory.
>
>
> **Q1**: Justification of limited effectiveness.
> **A1**: We want to stress that OOM generalization is still a very challenging task where we have 1) no supervision at all and 2) no prior relationship between IM and OOM data.
> - Therefore, the baseline "aligned" is for reference where we have numerous data with full supervision.
> - Directly comparing our method with "aligned" would be inappropriate.
> - Although the gap is discouraging, this gives us opportunities for future research. We believe when closing such a gap, we would be approaching AGI.
>
>
> **Q2**: Comparisons with SOTA unsupervised uni-modal methods.
> **A2**: Thanks, since an unsupervised method that is modality-agnostic is hard to find, therefore, we employ a very competitive method Momentum Contrast (MoCo) as a baseline.
> - For the semi-supervised case, we use MoCo combined with EntMin to train OOM data.
> - For the unsupervised case, we directly employ MoCo. The result on MSR-VTT with R@1 is shown below:
>     | Setting | Method      |  Aud |  Lan |  Vid |
>     |---------|-------------|:----:|:----:|:----:|
>     | Semi    | MoCo+EntMin | 20.5 | 21.1 | 23.4 |
>     | Semi    | COX         | **23.3** | **23.4** | **26.5** |
>     | Unsup   | MoCo        | 10.5 | 10.2 | 12.6 |
>     | Unsup   | COX         | **13.5** | **16.5** | **15.2** |
>     - We can see that under both scenarios, the performance improvement of COX is still significant, which justifies the effectiveness of COX. We will incorporate such a baseline into our revision. If there are any other baseline methods to be incorporated, please let us know.

---

> > ### Comment · Reviewer_KNt4 · 2024-11-24
> > **Thanks for the reply**
> >
> > Thank you for your response! I still have two concerns:
> > 1.  Has the unsupervised method to be modality-agnostic? CoX should demonstrate comparable performance with sota baselines for each different modality. If not, why not just using SSL method?
> > 2. Can the proposed method enhance existing method? for example, can MoCo+CoX be better than MoCo on images? This can  make it more meaningful to extract knowledge from other modalities to enhance new one.

---

> ### Author Response · Authors · 2024-11-22
> **Further Discussion**
>
> Dear Reviewer KNt4,
>
> We really appreciate your acknowledgement of our contribution, which makes us feel it is our obligation to improve the quality of this paper as much as we can.
>
> Therefore, we are eager to know if there are any further questions or unclearity left. We would be glad to have further discussions if there are any suggestions or questions.
>
> Thank you again for your valuable comments, we couldn't have improved our paper without your help.
>
> Best,
> Authors.

---

> ### Author Response · Authors · 2024-11-24
> **Further Discussion**
>
> Dear reviewer KNt4,
>
> - Comparison to SOTA baselines in each uni-modality.
>     - Thanks for the suggestion, here we consider the two most well-studied modalities, vision and language, and choose two recent SOTA baselines for comparison.
>     - For vision, we choose FreeMatch which conducts balanced semi-supervised learning under the FixMatch framework.
>     - For language, we choose MixText which conducts mixup augmentation for semi-supervised training.
>     - Here we employ them under the OOM generalization setting, and show the result below:
>         | Vision   | NYU-D | Language | MSRVTT |
>         |----------|:-----:|----------|:------:|
>         |FreeMatch |  45.2 | MixText  |  21.2  |
>         | COX      |  **52.3** | COX      |  **23.4**  |
>         - Even though the two baselines were effective under their original setting, their performance is still limited when applied to a challenging multimodal dataset with scarce knowledge. As we can see, COX still shows very effective performance compared to them, again justifying the benefits of leveraging IM data.
>
> - Ablation of combining COX for vision tasks.
>     - Thanks a lot for the suggestion, it is indeed an important benefit of the OOM generalization study.
>     - Here we show the result below:
>         | Setting | Method      |  NYU-D  |
>         |---------|-------------|:-------:|
>         | Unsup   | MoCo        |  15.7   |
>         | Unsup   | MoCo+COX    |**23.8** |
>         - Delightfully, we observe significant performance improvement on RGB of NYU-D dataset, this is because COX unleashes the potential label information from IM data to enhance label prediction of OOM, i.e., RGB here.
>         - Such a finding implies that COX can extract knowledge from other modalities to enhance new ones, which is the main goal of this study.
>
>
> Thank you again for your valuable and insightful opinions, we are really glad to achieve some additional findings that justify our method. We will carefully conduct experiments to enrich the final version of this paper. Without your help and effort, we couldn't reach this achievement.
>
> Kind regards,
> Authors.

---

> > ### Comment · Reviewer_KNt4 · 2024-11-25
> > **Thanks for the reply**
> >
> > Thank you for your response and some of my concerns have been addressed. I will consider raise the score.

---

> > > ### Author Response · Authors · 2024-11-25
> > > **Thank you for your encouragement**
> > >
> > > Dear reviewer KNt4,
> > >
> > > We cannot show our gratitude enough for your insightful suggestions. Even though our intention is mainly to clarify misunderstandings and improve our paper, we are really glad that our effort is recognized by your affirmation and willingness to increase scores.
> > >
> > > If you still have further questions or suggestions, please don't hesitate to let us know. Thank you so much.
> > >
> > > Best regards,
> > > Authors.

---

> ### Author Response · Authors · 2024-12-03
> **Anything Else We Can Help?**
>
> Dear reviewer KNt4,
>
> Thank you for your previous recognition of your contribution, we are really glad that our effort has helped address your concerns. Moreover, we have carefully revised our paper with the following aspects based on your suggestions:
>
> - Incorporating additional baseline MoCo in the main comparison. (Table 3)
> - Justifying the effectiveness of COX by showing its benefits under various correspondence levels. (Figure 7)
> - Testifying the superiority of COX over competitive uni-modal baselines. (Table 5)
> - Enhancing IM performance of MoCo by leveraging COX. (Table 6)
>
> Before the discussion ends within a day, we hope to know whether we can help with any additional concerns that affect your rating. If there are any additional suggestions or modifications that you would recommend, please don't hesitate to let us know.
>
> Again we are deeply grateful for your help and taking your valuable time, looking forward to hearing from you soon.
>
> Sincerely,
> Authors.

---

### Official Review · Reviewer_oDE6 · 2024-10-27

**Soundness:** 2
**Presentation:** 2
**Contribution:** 2
**Rating:** 6
**Confidence:** 4

**Summary:**

This paper addresses the problem of out-of-model (OOM) generation and proposes a novel technique, COX. The authors perform experiments on five datasets, comparing their method with several baseline approaches.

**Strengths:**

The problem statement is clear.

**Weaknesses:**

Insufficient and weak baselines: I am especially concerned the proposed approach's performance compared with SSL.

Some notations should be improved, and a few detailed explanations should be added.

**Questions:**

From my understanding, the authors consider a scenario where the training data consists of two parts: IM, where both inputs and labels are available, and OOM, where only inputs are provided without any labels (unsupervised case), and only a limited amount of labeled data is present. Please correct me if I’m mistaken, but it seems that no data contains both IM and OOM inputs. In other words, there is no data where the pair (x1, x2)—representing two modalities—is known.

If the above assumption is correct, how do we sample from the joint distribution p(xO | xI)p(xI) in Equation (6)?

I find the assumption stated below Equation (1) too strong. It implies that the proposed method only applies when xO can be directly inferred from xI. For example, let’s assume a1 = a2 = a, representing the sound of a dog barking, and b1 and b2 represent images of a golden retriever and a border collie, respectively. V is a constant vector, and Y represents the class "golden retriever." While it is true that P(V|xI=a1)P(Y|xI=a1) = P(V|xI=a2)P(Y|xI=a2), given that a1 = a2, we see that P(V,Y=is_golden_retriever|xO=b1, xI=a1) does not equal P(V,Y=is_golden_retriever|xO=b2, xI=a2).

In Line 255, do h1* and h2* refer to the classifiers corresponding to modality 1 and modality 2, respectively? If so, how can they be applied to xO, which belongs to a different modality?

The baselines should be reconsidered, especially by including stronger baselines. For instance, there are many state-of-the-art semi-supervised learning (SSL) approaches, and simply adding Gaussian perturbations to the input may not be sufficient. Stronger SSL baselines are crucial, as otherwise, it could be argued that using SSL alone is enough, which raises the question, why COX? Additionally, the 'Random' baseline is too simplistic, effectively leaving the unsupervised setting in Table 2 with only one baseline.

Furthermore, for the semi-supervised case, it is not surprising that the baseline performs worse, given the very limited amount of labeled data used for training.

I would like to bring [1] to the authors' attention, which could be helpful for better understanding multimodality:

[1] Zihui Xue et al., The Modality Focusing Hypothesis: Towards Understanding Crossmodal Knowledge Distillation, ICLR 2022.

---

> ### Author Response · Authors · 2024-11-21
> **Rebuttal**
>
> We thank reviewer oDE6 for your constructive suggestions. We have tried our best to answer all your questions, hope you find it satisfactory.
>
> **Q1**: Explanation of sampling $p(x^{\mathrm{O}}|x^{\mathrm{I}})p(x^{\mathrm{I}})$ without correspondence.
> **A1**: As discussed in [1], the VIB framework can be applied in an unsupervised manner by assigning each OOM data to one cluster centroid, thus sparing the need for correspondence.
> - In our practice, we conduct a warm-up phase to initialize the centroid using the modality agreement rule.
> - Further, the cluster is enhanced by learning class-wise mapping across modalities.
>
> [1] Alemi, A.A., Fischer, I., Dillon, J.V. and Murphy, K., 2016. Deep variational information bottleneck. arXiv preprint arXiv:1612.00410.
>
> **Q2**: Justification of assumption $p(V, Y|X^{\mathrm{O}}, X^{\mathrm{I}})=p(V|X^{\mathrm{I}})p(Y|X^{\mathrm{I}})$.
> **A2**: We want to stress that we consider the case where correspondent modalities are related to only one instance, and each instance is different from others, which is the same setting as CLIP and various multi-modal studies.
> - Therefore, the assumption ("$a1=a2=a$") proposed by the reviewer is quite different where two instances have the same modality data, which creates a many-to-one correspondence, but we consider one-to-one correspondent.
> - As a result, such an assumption introduces a population shift which violates our problem setting.
> - In our setting, it is sufficient to infer the labels $Y$ and latents $V$ by giving $X^{\mathrm{I}}$, because $X^{\mathrm{O}}$ is redundant. Hence, our assumption holds.
>
> **Q3**: Explanation of applying classifiers to different modalities.
> **A3**: Applying the classifier across different modalities is based on the modality connection.
> - Once we build the connection that maps IM data to OOM data, we can form the IM data in a feature space that is similar to OOM data.
> - Since IM data is fully known, we can obtain the near-optimal classifiers in such a space.
> - Therefore, the optimal classifiers can be applied to OOM data.
> - As we show in Figure 5: As connection closes the modality gap, applying classifiers from IM to OOM data achieves improved performance.
>
>
> **Q4**: Comparison with SOTA unsupervised uni-modal baselines.
> **A4**: Thanks, since an unsupervised method that is modality-agnostic is hard to find, therefore, we employ a very competitive method Momentum Contrast (MoCo) as a baseline.
> - For the semi-supervised cases, we use MoCo combined with EntMin to train OOM data.
> - For the unsupervised case, we directly employ MoCo. The result on MSR-VTT with R@1 is shown below:
>     | Setting | Method      |  Aud |  Lan |  Vid |
>     |---------|-------------|:----:|:----:|:----:|
>     | Semi    | MoCo+EntMin | 20.5 | 21.1 | 23.4 |
>     | Semi    | COX         | **23.3** | **23.4** | **26.5** |
>     | Unsup   | MoCo        | 10.5 | 10.2 | 12.6 |
>     | Unsup   | COX         | **13.5** | **16.5** | **15.2** |
>     - We can see that under both scenarios, the performance improvement of COX is still significant, which justifies the effectiveness of COX. We will incorporate such a baseline into our revision. If there are any other baseline methods to be incorporated, please let us know.
>
> **Q5**: Related works.
> **A5**: Thanks for providing such an interesting work, we have carefully included the reference and discussed its relationship with our work.

---

> ### Author Response · Authors · 2024-11-22
> **Further Discussion**
>
> Dear Reviewer oDE6:
>
> We are deeply grateful for your insightful comments. We have made a maximum effort to address each question by providing both theoretical and empirical justifications, such as:
> 1) applicability of VIB without correspondence,
> 2) practicality of our assumption,
> 3) explanation of applying classifiers across modalities,
> 4) comparison with SOTA baseline.
>
> We hope our answer could clarify your concerns. If any questions remain, please don't hesitate to let us know. We will be more than happy to have a detailed discussion. Thanks again for your valuable contribution to this paper, without which our paper couldn't have been further improved.
>
> Warm regards,
> Authors.

---

> ### Comment · Reviewer_oDE6 · 2024-11-22
> **Thanks for the reply**
>
> Thank you for your response! I have two follow-up questions: (i) The assumption that XO can be inferred from XI seems overly strong. Could you please clarify or elaborate on this? (ii) Could you explain what you mean by "an unsupervised method that is modality-agnostic is hard to find"?

---

> ### Author Response · Authors · 2024-11-22
> **Follow-up Discussion**
>
> Dear Reviewer oDE6,
>
> Thank you for your prompt reply. We are happy to address your remaining questions.
>
> - Assumption $p(V, Y|X^{\mathrm{O}}, X^{\mathrm{I}})=p(V|X^{\mathrm{I}})p(Y|X^{\mathrm{I}})$ that implies $X^{\mathrm{O}}$ can be inferred from $X^{\mathrm{I}}$.
>     - We want to clarify that such an assumption is not based on inferring $X^{\mathrm{O}}$ from $X^{\mathrm{I}}$.
>         - For example, given dog bark audio, we can know it corresponds to an instance dog. Similarly, given a dog image, we can know it is from a dog, too. So when given dog bark, we assume we can infer the label dog with or without the dog image.
>         - On the other hand, here we do not assume we can identify dog bark from a dog image.
>     - In our method, we use the VIB framework to connect $X^{\mathrm{O}}$ and $X^{\mathrm{I}}$, which are both observable.
>         - Theoretically, given two observable variables, it is feasible to map from one to another using VIB [1], including supervised and unsupervised scenarios [2]. Moreover, as stated in [3], it is reasonable to learn such mappings that connect different modalities.
>         - Practically, real-world applications from [4] conduct cross-modal translations to map across medical modalities such as magnetic resonance images (MRI) and electrocardiograms (ECGs), which again justifies our methodology.
>
> - Explanation of "an unsupervised method that is modality-agnostic is hard to find".
>     - Let's consider the three essential aspects of machine learning: data, model, and optimization:
>         - The limitation of finding a modality-agnostic method is mostly because of the data aspect, where many effective data-augmentation techniques cannot be applied to different modalities. Hence, many popular SSL methods based on image data are not applicable.
>         - Therefore, we consider MoCo because of two of its advantages:
>             - Momentum update of model,
>             - Contrastive learning for optimization,
>
>             which are still very effective for representation learning, thus the superiority of COX over MoCo can well justify its effectiveness. We will complete the baseline comparison with MoCo and update it to the revision soon.
>
> We deeply appreciate your follow-up discussion, we hope the above clarification can address your remaining concerns. If there are any further questions left, we are here to provide answers with our maximum effort.
>
>
>
> [1] Alemi, A.A., Fischer, I., Dillon, J.V. and Murphy, K., 2016. Deep variational information bottleneck. arXiv preprint arXiv:1612.00410.
> [2] Slonim, N., Atwal, G.S., Tkačik, G. and Bialek, W., 2005. Information-based clustering. Proceedings of the National Academy of Sciences, 102(51), pp.18297-18302.
> [3] Lu, Z., 2023. A theory of multimodal learning. Advances in Neural Information Processing Systems, 36, pp.57244-57255.
> [4] Radhakrishnan, A., Friedman, S.F., Khurshid, S., Ng, K., Batra, P., Lubitz, S.A., Philippakis, A.A. and Uhler, C., 2023. Cross-modal autoencoder framework learns holistic representations of cardiovascular state. Nature Communications, 14(1), p.2436.

---

> ### Author Response · Authors · 2024-11-24
> **Follow-up Discussion**
>
> Dear Reviewer oDE6,
>
> We are grateful for your follow-up comments, we have tried our best to address all the remaining questions. We hope that our reply can clearly and effectively provide an explanation for our study. As the discussion phase is ending soon, we are eager to know whether our answer is satisfactory for you. If there are any other factors that affect your opinions about this paper, we hope you could kindly let us know.
>
> Again thank you for your help and effort, we sincerely acknowledge your contribution for sparing your valuable time to reply to us.
>
> Kind regards,
> Authors.

---

> ### Author Response · Authors · 2024-11-30
> **Your Further Opinions Are Required**
>
> Dear Reviewer oDE6,
>
> Thank you for your initial comments. However, we have not received any follow-up from you, and your opinion regarding our justification remains unclear. If our responses are unsatisfactory, we kindly request that you specify the concerns that remain unresolved, as reviewers are encouraged to provide detailed feedback.
>
> Given that your rating is still negative, we are curious why no additional questions have been raised, despite our careful efforts to address all your concerns.
>
> We look forward to hearing from you and deeply appreciate your time and effort in providing further justification for your rating.
>
> Sincerely,
> Authors

---

> ### Author Response · Authors · 2024-12-02
> **The Last Day of Providing Your Opinions**
>
> Dear reviewer oDE6,
>
> We thank you for reviewing this paper. However, we never received any further comments from you. So we could not make any progress with you if we do not know what is your opinion.
>
> Our discussions with all other reviewers have been successful, and we have achieved a consensus. We are still hoping to know whether we have effectively addressed your concerns since you are the last one. Hope we can have the chance to hear from you before the discussion ends in a day, and we would be deeply appreciated.
>
> We understand it has been a long time, so here we summarize our previous reply:
> - We provided intuitive justification for our assumption from both theoretical and practical aspects.
> - We explained our choice of baseline.
> - Moreover, we have provided additional empirical experiments by comparing COX to uni-modal methods. (Table 5)
>
> If you can spare a little time to take a look, it would be a big help for us. We just hope we can work together to ensure a fair judgment for this paper.
>
> Best wishes,
> Authors.

---

> > ### Comment · Reviewer_oDE6 · 2024-12-03
> > **Thanks for the authors' responses**
> >
> > I sincerely thank the authors' dedicated efforts to address my questions. I am not completely convinced by their explanation on Eq. (1) of the paper. I concur with the authors on the other points and will raise my score to 6.

---

> > > ### Author Response · Authors · 2024-12-03
> > > **Thank you for your response**
> > >
> > > Dear reviewer oDE6,
> > >
> > > Thank you so much for your recognition of our effort, we sincerely mean it. We will carefully take your suggestions to clearly formulate our assumption in Eq. (1) to make sure it is well-supported.
> > >
> > > Thanks again for raising the score, we will keep polishing this paper by considering all suggestions from the reviewers.
> > >
> > > Best wishes,
> > > Authors.

---

### Official Review · Reviewer_PuJV · 2024-10-29

**Soundness:** 3
**Presentation:** 3
**Contribution:** 2
**Rating:** 6
**Confidence:** 4

**Summary:**

This work focuses on the semi-supervised and  unsupervised scenarios of *Out-of-Model* (OOM) Generalization without given instance-level model correspondence. This work proposes an OOM generalization method based on the interactive relationship between modalities, *connect & explore* (COX), in an attempt to extract the commonality that can help partially comprehend OOM data based on IM data. It introduce a variational information bottleneck framework to connect OOM data and In-Model (IM) data and extract shared information. Finally, several sets of experiments demonstrate the effectiveness and extensibility of the proposed method.

**Strengths:**

This work focuses on the novel and practical issues in the field of *Out-of-Model* (OOM) generalization.

This work proposes an approach that attempts to combine common knowledge based on connections across modalities with unique knowledge of OOM data based on modality disagreement.

This work conducts multiple sets of experiments to verify the effectiveness and extensibility of the proposed method.

**Weaknesses:**

(i) The OOM problem has also been discussed in other works, such as IB for modality missing, cross-modal generalization, etc., ([1-5] for example) but not mentioned in the paper (nor experiments). Please explain the difference between this paper and these works. [1] Unpaired image-to-speech synthesis with multimodal information bottleneck. [2] Visual explanations of image-text representations via multi-modal information bottleneck attribution. [3] Dynamic Multimodal Information Bottleneck for Multimodality Classification. [4] SimMMDG: A simple and effective framework for multi-modal domain generalization

(ii) The work mentions "prediction uncertainty" in section 4.2, but after carefully reviewing all the contents, it is hard to find the definition or description of prediction uncertainty, while without the direct description of this concept in the experiment section for verification.

(iii)The work mentions "the performance of COX is affected by the quality of IM perceptors", but only contrast the performance between ImageBind and LanguageBind in the experiment section can not fully illustrate this conclusion.

(iv) Has some loopholes, e.g.,

  - Tense and grammatical errors, e.g., "OMM data" -> "OOM data"
  - Confusing notations, e.g.,"given IM data *X<sup> O </sup>*and OOM data *X<sup> I </sup>*"

Please correct the grammatical mistakes and polish them if possible.

(v) In the experimental part, the results in Table 2 and Table 3 can not adequately illustrate this conclusion "leveraging the knowledge from IM perceptors can indeed help OOM generalization compared to using OOM data alone". How do the results in Table 2 and Table 3 show the performance improvement of the proposed method compared to the methods using only OOM data?

**Questions:**

Please see 'weakness', which simply can be summarised as:

(i) What the advantages of this work vs. previous works?

(ii) What is "prediction uncertainty" described in the article and how is it reflected?

(iii) Can the conclusion that COX performance is affected by quality of IM perceptors be fully explained based on the final performance results of only two IM perceptors, ImageBind and LanguageBind? What are the quality differences between ImageBind and LanguageBind reflected?

(iv) How do the data in Table 2 and Table 3 reflect that leveraging the knowledge from IM perceptors can indeed help OOM generalization compared to using OOM data alone?

(v) There are some grammatical errors and confusing notations in the article. Can it be further polished and revised?

---

> ### Author Response · Authors · 2024-11-21
> **Rebuttal**
>
> We appreciate reviewer PuJV for your positive opinions. Here we carefully take all your suggestions and address all concerns.
>
>
> **Q1**: Discussion on related works.
> **A1**: Thanks for pointing out. We have carefully discussed the difference between them and our work. Please see the revised paper.
>
>
> **Q2**: Definition of prediction uncertainty.
> **A2**: Thanks, we have stressed this denotion in the revision: The "prediction uncertainty" denotes the data with uncertain prediction, i.e., low accuracy.
>
> **Q3**: Relationship between COX performance and IM perceptor qualities.
> **A3**: Here we explain and justify our claim:
> - The quality difference between ImageBind and LanguageBind is they use different modalities for binding and have different parameter scales.
> - To further justify this claim, we leverage various CLIP as IM perceptors for RGB and Language modalities, and show the OOM performance on Depth data using the NYU-D dataset:
>     |     | ViT-S/16 | ViT-B/16 | ViT-L/16 |
>     |-----|:--------:|:--------:|:--------:|
>     | COX |   46.4   |   50.9   |   52.4   |
>     - We can find that the performance of COX is dependent on the employed IM perceptors, therefore, it is essential to enhance IM perceptors to employ COX.
>
>
>
> **Q4**: Typos.
> **A4**: Thanks for pointing out, we have carefully refined our paper and polished all unclear notations.
>
>
> **Q5**: Justification of "IM perceptors can help OOM generalization".
> **A5**: In our comparison, the baselines such as ERM, SSL, and EntMin only learn from OOM data without any IM knowledge. Contrastively, our method leverages the knowledge from IM data:
> - Achieving superior performance than baselines.
> - Enabling unsupervised OOM generalization without any correspondence.
>
> To further provide intuitive justification, we show the benefits of COX by varying the number of correspondence in OOM data on MSR-VTT:
> - We use ERM and EntMin as baselines, and vary the correspondence number to 90%, 60%, 30%, and 10%, as shown below:
>     | Method   |  90% |  60% |  30% |  10% |
>     |----------|:----:|:----:|:----:|:----:|
>     | ERM      | 68.4 | 60.1 | 46.9 | 38.2 |
>     | EntMin   | 69.2 | 61.6 | 48.5 | 39.4 |
>     | COX      | 69.4 | 63.9 | 55.1 | 48.8 |
>     | $\Delta$ |  0.2 |  2.3 |  5.6 |  9.6 |
>     - First of all, we observe that COX brings more benefits when correspondence is more scarce.
>     - This is because sufficient correspondence can maximally uncover the knowledge of OOM data. As correspondence gets less, the knowledge that can be explored from correspondence decreases.
>     - However, COX leverages the knowledge from IM data which brings more benefits compared to using OOM data alone.

---

> ### Author Response · Authors · 2024-11-22
> **Further Discussion**
>
> Dear Reviewer PuJV:
>
> We really appreciate your insightful opinions that greatly helped us improve this paper. We carefully took every single question and tried our best to answer them, including related works and study on IM perceptor quality. If there are any concerns unresolved, we would be glad to have further discussions.
>
> Thanks again for taking your valuable time and contribution on this paper. Even though it is anonymous, we want you to know that you are deeply appreciated. We are looking forward to hearing from you soon.
>
> Sincerely,
> Authors.

---

### Official Review · Reviewer_Zg1q · 2024-11-02

**Soundness:** 2
**Presentation:** 2
**Contribution:** 2
**Rating:** 6
**Confidence:** 4

**Summary:**

This paper explores a novel model generalization problem, specifically focusing on out-of-modal generalization in the absence of label and modality correspondence. The authors employ a two-stage generalization approach, first constructing modality-shared information and then focusing on modality-specific information. They propose different training losses for both semi-supervised and completely unsupervised scenarios.

**Strengths:**

1.The problem addressed in the paper is quite novel, and the semi-supervised scenario provides valuable insights."

2.Extensive experiments are conducted to demonstrate the effectiveness of the proposed method.

**Weaknesses:**

1.The paper contains several typos that affect readability and need to be corrected. Regarding line 195, it seems that the symbols for IM data $ X^O$ and OOM data $ X^I$ are used incorrectly. Also in Eq.3, why is there an integral sign before $p ( X^O, X^I ) $

2.The proof in the paper lacks rigor. How is the lower bound derived from Equation 3 to Equation 4? Although the proof draws on VIB, there are still significant discrepancies.

3.Similarly, the proof in Appendix A.1 for Equation 20, which relies on the non-negativity of the KL divergence, is not sufficiently rigorous.

4.In Section 3.2, there is a lack of necessary descriptions regarding the trained model, such as the input and output of the decoder $q( X^O, X^I)$.

**Questions:**

1.Does the inclusion of $L_{con}$ with label $ y $ imply that exploring connections across modalities requires the use of samples with correspondence for training?

2.In a completely unsupervised setting, the feature distributions of different modalities may differ significantly. Is the choice of the anchor method reasonable in this context?

3.Could you provide a more detailed proof regarding the upper and lower bounds in Sec 3.2?

4.In the ablation experiments, the results of removing any module are significantly lower than those of EntMin. Would applying these two losses individually on EntMin lead to improvements?

---

> ### Author Response · Authors · 2024-11-21
> **Rebuttal**
>
> We thank reviewer Zg1q for your valuable opinions. We have carefully addressed all your comments and hope that you will find our response satisfactory.
>
> **Q1**: Typos and mathematical formulations.
> **A1**: Thanks for pointing out, we have carefully clarified all mathematical formulations:
> - IM data $X^{\mathrm{I}}$ and OOM data $X^{\mathrm{O}}$;
> - Eq. 3:
> \begin{align}
>     I(X^{\mathrm{O}}; X^{\mathrm{I}})&=\int dx^{\mathrm{O}}dx^{\mathrm{I}} p(x^{\mathrm{O}}, x^{\mathrm{I}})\log\frac{p(x^{\mathrm{O}}, x^{\mathrm{I}})}{p(x^{\mathrm{O}})p(x^{\mathrm{I}})}=\int dx^{\mathrm{O}}dx^{\mathrm{I}} p(x^{\mathrm{O}}, x^{\mathrm{I}})\log\frac{p(x^{\mathrm{O}}|x^{\mathrm{I}})}{p(x^{\mathrm{O}})},
> \end{align}
>
> For details, please check the revised paper.
>
> **Q2**: Explanation of lower-bound derivation from Eq. 3 to Eq. 4.
> **A2**:
> In Eq. 3, we first replace $p(x^{\mathrm{O}}|x^{\mathrm{I}})$ with $\int dv p(x^{\mathrm{O}}|v)p(v|x^{\mathrm{I}})$, where $p(v|x^{\mathrm{I}})$ can be modeled by the IM perceptors, and $p(x^{\mathrm{O}}|v)$ can be esitmated via a decoder $q(x^{\mathrm{O}}|v)$, then we have:
> \begin{align}
> \int dx^{\mathrm{O}}dx^{\mathrm{I}} p(x^{\mathrm{O}}, x^{\mathrm{I}})\log\frac{\int dv p(x^{\mathrm{O}}|v)p(v|x^{\mathrm{I}})}{p(x^{\mathrm{O}})}=
> \int dx^{\mathrm{O}}dx^{\mathrm{I}}dv p(x^{\mathrm{O}}, x^{\mathrm{I}})\log\frac{p(x^{\mathrm{O}}|v)}{p(x^{\mathrm{O}})/p(v|x^{\mathrm{I}})}
> \end{align}
> Then, based on $\int dx^{\mathrm{O}} p(x^{\mathrm{O}}|v)\log p(x^{\mathrm{O}}|v)\ge \int dx^{\mathrm{O}} p(x^{\mathrm{O}}|v)\log q(x^{\mathrm{O}}|v)$, and since the denominator stays the same, we can follow the same derivation as VIB and obtain:
> $$
> \int dx^{\mathrm{O}}dx^{\mathrm{I}}dv p(x^{\mathrm{O}}, x^{\mathrm{I}})\log\frac{p(x^{\mathrm{O}}|v)}{p(x^{\mathrm{O}})/p(v|x^{\mathrm{I}})}\ge\int dx^{\mathrm{O}}dx^{\mathrm{I}}dv p(x^{\mathrm{O}}, x^{\mathrm{I}})\log\frac{q(x^{\mathrm{O}}|v)}{p(x^{\mathrm{O}})/p(v|x^{\mathrm{I}})},
> $$ which is our Eq. 4.
>
>
> **Q3**: Relying on the non-negativity of KL divergence.
> **A3**: Based on the definition of KL divergence, its non-negativity is a fact. Relying on such property for derivation is sufficient, as done by all VIB frameworks. If there are any other unclear derivations, please let us know, thanks.
>
> **Q4**: Details description of the training process.
> **A4**: There are two parts in Eq. 8:
> - The first part aims to reconstruct OOM data by applying a decoder $q(X^{\mathrm{O}}|V)$ on top of the IM perceptors, together they model $p(X^{\mathrm{O}}|X^{\mathrm{I}})$.
>     - For training, we optimize the parameters of both the decoder and IM perceptors to maximize the log-likelihood $\log q(X^{\mathrm{O}}|X^{\mathrm{I}})$.
>     - Compared to the original VIB, the KL-divergence of the latent variables is dropped because we employ pre-trained models that already possess latent structure.
> - The second part aims to harness the label information of IM data to enhance the label prediction of OOM data.
>     - We can derive a KL-divergence between label predictions of IM data and OOM data: $\text{KL}(p(Y|X^{\mathrm{I}})\|h^{\mathrm{O}}(Y|X^{\mathrm{O}}))$.
>     - This KL divergence implies that we can take advantage of the class information of IM data to explore OOM data, as done by our modality agreement criterion.
>     - Note that we do not parameterize the latent variable $V$ here, so $p(X^{\mathrm{O}}|X^{\mathrm{I}})$ is not modeled, thus fixed.
>
> We have added more details of the training process in the revision, please check the revised paper.
>
> **Q5**: Necessity of correspondence for training VIB for including label $y$.
> **A5**: Accessing correspondence indeed simplifies the problem, as discussed in the semi-supervised scenario. However, in the unsupervised scenario without correspondence, we showed that class information alone is sufficient for modality connection:
> - We first explore the OOM data with commonality via the modality agreement rule.
> - Then, initial clustering can be formed to produce class centroids.
> - Practically, we use centroids to map from IM data to OOM data.
> - Theoretically, the unsupervised VIB as discussed in [1, 2] can be applied, which also enforces each OOM data to be assigned to one cluster centroids.
>
> [1] Alemi, A.A., Fischer, I., Dillon, J.V. and Murphy, K., 2016. Deep variational information bottleneck. arXiv preprint arXiv:1612.00410.
> [2] Slonim, N., Atwal, G.S., Tkačik, G. and Bialek, W., 2005. Information-based clustering. Proceedings of the National Academy of Sciences, 102(51), pp.18297-18302.

---

> > ### Author Response · Authors · 2024-11-21
> > **Rebuttal**
> >
> > **Q6**: Justification of anchor method under significant distribution difference.
> > **A6**: The anchor point method aims to initialize the feature space, which is not required to be perfect.
> > - The subsequent modality connection can further close the distribution difference to match IM data and OOM data.
> > - Our VIB framework also leverages the information contained in OOM data for clustering.
> > - To further justify the effectiveness of our anchor point, we compare the accuracies of anchor points and non-anchor points, and use the unsupervised method SSL as a baseline, as shown below:
> >     |            | MSR-VTT | NYU-D |
> >     |------------|:-------:|:-----:|
> >     | Anchor     |**31.7**|**42.1**|
> >     | Non-Anchor |   6.2   |  8.0  |
> >     | SSL        |   10.1  |  13.6 |
> >     - We observe that the selected anchor points show significantly better accuracy than non-anchor points, thus our anchor point method is reasonable under unsupervised scenarios.
> >
> > **Q7**: Detailed proof of lower bound of VIB.
> > **A7**: We have provided detailed proof of the lower-bound in **Q2**. Here we provide details of deriving upper-bound:
> > \begin{align}
> >     &I(X^{\mathrm{O}}; X^{\mathrm{I}}|Y)=\int dx^{\mathrm{O}}dx^{\mathrm{I}}dy p(x^{\mathrm{O}}, x^{\mathrm{I}}, y)\log\frac{p(x^{\mathrm{O}}, x^{\mathrm{I}}|y)}{p(x^{\mathrm{O}}|y)p(x^{\mathrm{I}}|y)}.
> > \end{align}
> > Based on Bayes' rule, we have $p(x^{\mathrm{O}}, x^{\mathrm{I}}|y)=\frac{p(x^{\mathrm{O}}, x^{\mathrm{I}}, y)}{p(y)}$, $p(x^{\mathrm{O}}|y)=\frac{p(y|x^{\mathrm{O}})p(x^{\mathrm{O}})}{p(y)}$, and $p(x^{\mathrm{I}}|y)=\frac{p(y|x^{\mathrm{I}})p(x^{\mathrm{I}})}{p(y)}$, replacing back to above equation, we have
> > \begin{align}
> > I(X^{\mathrm{O}}; X^{\mathrm{I}}|Y)=&\int dx^{\mathrm{O}}dx^{\mathrm{I}}dy p(x^{\mathrm{O}}, x^{\mathrm{I}}, y)\log\frac{p(x^{\mathrm{O}}, x^{\mathrm{I}}, y)p(y)}{p(y|x^{\mathrm{O}})p(y|x^{\mathrm{I}})p(x^{\mathrm{O}})p(x^{\mathrm{I}})}.
> > \end{align}
> > Therefore, after integral on the joint distribution $p(x^{\mathrm{O}}, x^{\mathrm{I}}, y)$, we can obtain the entropy terms: $-H(Y)$, $H(Y|X^{\mathrm{I}})$, $H(X^{\mathrm{O}})$, and $H(X^{\mathrm{I}})$, which are ignored from model training, and having the following left:
> > $$
> > \int dx^{\mathrm{O}}dx^{\mathrm{I}}dy p(x^{\mathrm{O}}, x^{\mathrm{I}}, y)\log\frac{p(x^{\mathrm{O}}, x^{\mathrm{I}}, y)}{p(y|x^{\mathrm{O}})}.
> > $$
> > Further, we factorize $p(x^{\mathrm{O}}, x^{\mathrm{I}}, y)=p(y|x^{\mathrm{I}})p(x^{\mathrm{O}}|x^{\mathrm{I}})p(x^{\mathrm{I}})$, and use a classifier $h(y|x^{\mathrm{O}})$ to estimate $p(y|x^{\mathrm{O}})$. Therefore, the above formulation is less than or equal to (because the estimation is the denominator):
> > $$
> > \int dx^{\mathrm{O}}dx^{\mathrm{I}}dy p(x^{\mathrm{O}}, x^{\mathrm{I}}, y)\log p(y|x^{\mathrm{I}})p(x^{\mathrm{O}}|x^{\mathrm{I}})p(x^{\mathrm{I}}) - \log h(y|x^{\mathrm{O}}),
> > $$ which completes the upper bound.
> >
> >
> > **Q8**: Ablation results compared to EntMin.
> > **A8**: We want to argue that directly comparing each module of COX to baselines is not reasonable because each of them only considers part of OOM data.
> > - Connection only focuses on data with commonality.
> > - Exploration only focuses on data with uniqueness.
> > - But EntMin is applied to all OOM data.
> > - If we combine the modules of COX with EntMin, the performance is highly possible to be better, however:
> >     - It is orthogonal to our research problem, which is leveraging IM data to explore OOM data.
> >     - It makes the methodology redundant, like naively combining two methods.

---

> ### Author Response · Authors · 2024-11-22
> **Further Discussion**
>
> Dear Reviewer Zg1q,
>
> We want to express our appreciation for your valuable suggestions, which greatly helped us improve the quality of this paper. We are also glad that you think our work is novel and insightful. We have made our maximum effort to address all your concerns about detailed proof, non-negativity assumption, and method effectiveness.
>
> Your further opinions are very important for evaluating our revised paper and we are hoping to hear from you soon. Thank you again for your effort and constructive opinions.
>
> Best,
> Authors.

---

> ### Author Response · Authors · 2024-11-23
> **Any Remaining Questions?**
>
> Dear Reviewer Zg1q,
>
> We have carefully taken all your previous comments into consideration, which are very constructive and helpful. Your effort and contribution to this paper is deeply appreciated.
>
> After careful formulation and reply, we have provided:
> - details of our derivation,
> - justification of our assumption,
> - description of the training process,
> - empirical analysis of the anchor point method.
>
> We are eager to know whether our reply could be helpful and want you to know that we are here to make our maximum effort to improve our paper. Thank you again for taking your time and nothing would be more delightful to hear your further opinions.
>
> Sincerely,
> Authors.

---

> ### Author Response · Authors · 2024-11-24
> **Discussion Phase Ending**
>
> Dear Reviewer Zg1q,
>
> We again thank you for the effort you put into reviewing. We understand that you might be busy at this time, and we would deeply appreciate it if you could spare a little time to take a look at our rebuttal.
>
> We all hope that the interactive discussion is effective so that there is no misunderstanding or unclarity left between the reviewers and authors. So we cherish this opportunity to discuss with you and have sufficient communication before a final decision is made.
>
> Since the discussion phase is ending, we sincerely hope we can hear from you soon so that we can provide our best effort and help to address your questions.
>
> Kind regards,
> Authors.

---

> > ### Comment · Reviewer_Zg1q · 2024-11-26
> > **Responses**
> >
> > Thanks for the responses. This paper is interesting, while I have a certain concern about the implementation. It will be better for authors to public the implementation details of the work, like other submissions, encompassing source code and implementation settings.

---

> > > ### Author Response · Authors · 2024-11-27
> > > **Public Source Code**
> > >
> > > Dear Reviewer Zg1q,
> > >
> > > Thank you for your reply and finding our paper interesting. This is a novel field with unimaginable potential, and we are trying our best to push the limit of generalization.
> > >
> > > Here we provide our main implementation and learning framework using ImageBind on MSR-VTT: https://anonymous.4open.science/r/Out-of-Modal-Generalization-1301.
> > > We will carefully integrate all backbone models and datasets into one framework soon if this paper is accepted.
> > >
> > > For any further questions about the implementation details that concern you, please feel free to let us know. We will be deeply appreciated to hear from you soon.
> > >
> > > Best wishes,
> > > Authors.

---

> > > > ### Comment · Reviewer_Zg1q · 2024-11-29
> > > > **Response to authors**
> > > >
> > > > Thanks for the responses. The source code is very helpful, and I will re-run the code for further empirical check. In additional, the responses still have a few issues, e.g., In A2 to Q2, the integral sign is directly moved out of log. A detailed description would be helpful.

---

> > > > > ### Author Response · Authors · 2024-11-30
> > > > > **Further Discussion**
> > > > >
> > > > > Dear reviewer Zg1q,
> > > > >
> > > > > Thank you for your feedback! We are delighted that you find our code helpful. Please feel free to reach out if you have any additional questions.
> > > > >
> > > > > Regarding the derivation in **A2**, we clarify that moving the integral $\int$ outside of $\log$ relies on Jensen's inequality. We have rectified the equality as follows:
> > > > > \begin{align}
> > > > > \int dx^{\mathrm{O}}dx^{\mathrm{I}} p(x^{\mathrm{O}}, x^{\mathrm{I}})\log\frac{\int dv p(x^{\mathrm{O}}|v)p(v|x^{\mathrm{I}})}{p(x^{\mathrm{O}})}\ge
> > > > > \int dx^{\mathrm{O}}dx^{\mathrm{I}}dv p(x^{\mathrm{O}}, x^{\mathrm{I}})\log\frac{p(x^{\mathrm{O}}|v)}{p(x^{\mathrm{O}})/p(v|x^{\mathrm{I}})}
> > > > > \end{align}
> > > > > The subsequent derivation remains unchanged.
> > > > >
> > > > > We truly appreciate your comment and believe it has prompted us to enhance the clarity and rigor of our derivation.
> > > > >
> > > > > We hope this addresses your concerns. Should you have any further questions, we are happy to assist and will respond promptly.
> > > > >
> > > > > Best regards,
> > > > > Authors

---

> > > > > > ### Comment · Reviewer_Zg1q · 2024-12-02
> > > > > > **Responses**
> > > > > >
> > > > > > Thank you for the response. The additional explanations are helpful, please make sure to incorporate them into the revised version. I will raise my score above the accepted line with 6.

---

> > > > > > > ### Author Response · Authors · 2024-12-02
> > > > > > > **Thanks For Your Support**
> > > > > > >
> > > > > > > Dear reviewer Zg1q:
> > > > > > >
> > > > > > > Thank you for your effort in reviewing and improving this paper, we will carefully revise this paper by considering all the valuable suggestions. We are also delighted that our explanation and justification have been helpful, and we sincerely appreciate your support.
> > > > > > >
> > > > > > > Best wishes,
> > > > > > > Authors.

---

### Official Review · Reviewer_2YMo · 2024-11-04

**Soundness:** 2
**Presentation:** 3
**Contribution:** 3
**Rating:** 6
**Confidence:** 4

**Summary:**

This paper proposes a novel method for Out-of-Modal (OOM) generalization, which uses a COX framework to achieve cross-modal knowledge transfer and generalization to unknown modalities. At the same time, this paper considers both semi-supervised and unsupervised scenarios. Experiments show that this method has a certain level of effectiveness.

**Strengths:**

1. The motivation behind this approach is clear, and studying out-of-model (OOM) generalization is meaningful.
2. The authors provide a detailed description of the proposed method, making it easy to follow and reproduce.

**Weaknesses:**

1. In the theoretical analysis, the paper uses $h^*$ to represent ideal optimal classifiers. However, in practice, it is difficult to find the optimal classifier. Although the authors compared the performance of ImageBind and LanguageBind, they should conduct more experiments to demonstrate the impact of using classifiers with different levels of accuracy on the final results.
2. The paper does not theoretically or experimentally explain why using the IM Perceptor is more effective than directly performing semi-supervised or unsupervised training.
3. If using the IM Perceptor indeed improves training results as mentioned in the issues above, then for the same task, would knowing more modalities enhance the generalization of OOM?
4. The paper contains numerous writing errors, such as "OMM" in line 494. Additionally, the logic of symbol definitions in the mathematical proof section is confusing, for example, using $X^O$ to represent IM data and $X^I$ to represent OOM data.

**Questions:**

Please see Weaknesses.

---

> ### Author Response · Authors · 2024-11-21
> **Rebuttal**
>
> We thank Review 2YMo for your constructive comments. Here we address all your concerns with careful justifications.
>
> **Q1**: 1) Practical assumption for optimal classifiers; 2) Experiments on classifiers with varied accuracy levels.
> **A1**: Thanks for the suggestion.
> Since we have sufficient labeled IM data, it is possible to achieve near-optimal classifiers:
> - Like most studies in AI/ML, we rely on ERM as the theoretical guarantee to approximate the optimal hypothesis class.
> - The IM percetors have been pre-trained on numerous data, further fine-tuned on well-known multi-modal datasets.
> - Therefore, assuming the IM perceptors are near-optimal classifiers is reasonable.
>
> Still, we acknowledge that real-world applications could suffer from low-data problems where we have both limited IM and OOM data:
> - To answer the question, we conduct experiments on MSR-VTT by using IM perceptors fine-tuned with different numbers of IM data, and show the result below:
>     |        |  10% |  40% |  70% | 100% |
>     |--------|:----:|:----:|:----:|------|
>     |Aud Acc.| 65.5 | 72.8 | 81.6 | 89.4 |
>     |Lan Acc.| 70.2 | 77.8 | 86.6 | 92.3 |
>     | Vid (OOM)| **18.4** | **28.9** | **36.7** | **48.8** |
>     - We can see that the OOM performance is significantly affected by the accuracy level of IM perceptors.
>     - Therefore, improving the performance of IM perceptors is vital for OOM generalization using COX.
> - We also emphasize that our goal is generalizing to rare OOM data without correspondence or labels, therefore, such a challenging situation is orthogonal to our problem.
>
> **Q2**: The reason for performance improvement by using IM perceptors.
> **A2**: Thanks. Here we justify both theoretical and experimental aspects:
> - Theoretically, we showed that the knowledge of one modality contains "commonality" and "uniqueness".
>     - Without correspondence or labels, the exploration of both commonality and uniqueness is largely hindered, leading to poor results.
>     - Through our VIB-based modality connection, we can maximally discover the commonality shared across modalities, providing essential guidance to understand OOM data.
>     - Further, COX leverages the guidance to explore uniqueness using semi-supervised techniques, thus surpassing baselines that neglect such guidance.
> - Experimentally, we vary the number of correspondence in OOM data from MSR-VTT to justify the benefits brought by COX:
>     - We use ERM and EntMin as baselines, and vary the correspondence number to 90%, 60%, 30%, and 10%, as shown below:
>         | Method   |  90% |  60% |  30% |  10% |
>         |----------|:----:|:----:|:----:|:----:|
>         | ERM      | 68.4 | 60.1 | 46.9 | 38.2 |
>         | EntMin   | 69.2 | 61.6 | 48.5 | 39.4 |
>         | COX      | 69.4 | 63.9 | 55.1 | 48.8 |
>         | $\Delta$ |  0.2 |  2.3 |  5.6 |  9.6 |
>         - First of all, we observe that COX brings more benefits when correspondence is more scarce.
>         - This is because sufficient correspondence can maximally uncover the knowledge of OOM data. As correspondence gets less, the knowledge that can be explored from correspondence decreases.
>         - However, COX leverages the knowledge from IM data which brings more benefits even with less correspondence.
>
> **Q3**: Study on enhancing OOM generalization by increasing modalities.
> **A3**: Thanks for the suggestion, here we conduct experiments on MSR-VTT by increasing the number of IM modalities as 0, 1, and 2:
> - For 0 IM modalities, we have the basic ERM method, for 1 IM modality, we conduct COX based on the prediction confidence instead of modality disagreement criterion, and for 2 IM modalities, we have the standard COX.
> - The result is shown below:
>     |                 |   0  |   1  |   2  |
>     |-----------------|:----:|:----:|:----:|
>     | OOM performance | 38.2 | 40.5 | 48.8 |
>     - We can see that as the number of IM modalities increases, the OOM performance also improves.
>     - Therefore, we can justify that using more IM modality data can benefit the OOM performance.
>     - Although well-known multi-modal datasets only have limited modality numbers due to the expense of acquiring modalities, using 3 modalities can provide insights for OOM generalization. We plan to discover the scaling performance on more modalities in the future.
>
> **Q4**: Typos.
> **A4**: Thanks for pointing out, we have carefully refined our paper and eliminated all the typos.

---

> > ### Comment · Reviewer_2YMo · 2024-11-24
> >
> > I appreciate the author's detailed response. I will consider slightly increasing the score because some of my concerns are being relieved. By the way, will the code be released after the acceptance?

---

> > > ### Author Response · Authors · 2024-11-24
> > > **Further Discussion**
> > >
> > > Dear Reviewer 2YMo,
> > >
> > > We appreciate your time and follow-up discussion. We are really glad to hear that our rebuttal helped address your concerns and that you are willing to increase your rating. Your insightful opinions indeed helped us a lot. For any other followed discussion, we are still more than happy to engage.
> > >
> > > Moreover, we assure you that we will carefully organize our source code and make it public to advance OOM generalization.
> > > Thanks again for your effort.
> > >
> > > Kind regards,
> > > Authors.

---

> ### Author Response · Authors · 2024-11-22
> **Further Discussion**
>
> Dear reviewer 2YMo,
>
> We really appreciate your efforts to help improve this paper. We have carefully addressed all the mentioned concerns, such as the analysis of the IM perceptors and the effect of modality numbers.
>
> Having further discussions really helps to achieve consensus and clarify misunderstandings, so we are eager to know if are there any remaining questions. We are more than happy to try our best to address them.
>
> Best,
> Authors.

---

> ### Author Response · Authors · 2024-11-23
> **Any Remaining Questions?**
>
> Dear Reviewer 2YMo,
>
> We want to show our gratitude again for your contribution to this paper, which has given us many insightful suggestions to improve our paper. We are looking forward to hearing from your further opinions about our rebuttal. We cherish the opportunity to polish our paper through this interactive discussion and hope you can spare a little time to take a look.
>
> If there are any further questions left, please feel free to let us know. We sincerely hope we could help clarify any further misunderstandings. Thank you again for your help and support, we truly appreciate it.
>
>
> Warm regards,
> Authors.

---

### Author Response · Authors · 2024-11-21
**General Response**

Dear Reviewers, AC, and SAC:

We deeply thank the hard work done by AC and SAC such as assigning reviewers, guiding the process, and further organizing the discussion. We also sincerely appreciate the reviewers for taking the time to read our paper, provide constructive opinions, and get involved in our discussion. Without your elaborative help, our paper could not have been improved.

We appreciate all the reviewers for their insightful and constructive reviews of our manuscript. Delightfully, we are glad that the reviewers found that:
- The proposed problem is **novel, insightful, and practical**. (Reviewers 2YMo, Zg1q, PuJV, and KNt4)
- Our research is **well-motivated, clearly-formulated, and easy-to-follow**. (Reviewers 2YMo, oDE6, and KNt4)
- The experimental evaluation is **effective, convincing, and extendable**. (Reviewers 2YMo, Zg1q, PuJV, and KNt4)

Based on all the comments from the reviewers, we provide a general response to the concerns raised by multiple reviewers. The individual responses are commented on below each review.

- Comparison with SOTA uni-modal baselines.
    - Since an unsupervised method that is modality-agnostic is hard to find, therefore, we employ a very competitive method Momentum Contrast (MoCo) as a baseline.
    - For the semi-supervised case, we use MoCo combined with EntMin to train OOM data.
    - For the unsupervised case, we directly employ MoCo. The result on MSR-VTT with R@1 is shown below:
        | Setting | Method      |  Aud |  Lan |  Vid |
        |---------|-------------|:----:|:----:|:----:|
        | Semi    | MoCo+EntMin | 20.5 | 21.1 | 23.4 |
        | Semi    | COX         | **23.3** | **23.4** | **26.5** |
        | Unsup   | MoCo        | 10.5 | 10.2 | 12.6 |
        | Unsup   | COX         | **13.5** | **16.5** | **15.2** |
        - We can see that under both scenarios, the performance improvement of COX is still significant, which justifies its effectiveness. We will incorporate such a baseline into our revision. If there are any other baseline methods to be incorporated, please let us know.

- Justification of "IM perceptors can help OOM generalization".
We vary the number of correspondences in OOM data from MSR-VTT to justify the benefits brought by COX.
    - We use ERM and EntMin as baselines, and vary the correspondence number to 90%, 60%, 30%, and 10%, as shown below:
        | Method   |  90% |  60% |  30% |  10% |
        |----------|:----:|:----:|:----:|:----:|
        | ERM      | 68.4 | 60.1 | 46.9 | 38.2 |
        | EntMin   | 69.2 | 61.6 | 48.5 | 39.4 |
        | COX      | 69.4 | 63.9 | 55.1 | 48.8 |
        | $\Delta$ |  0.2 |  2.3 |  5.6 |  9.6 |
        - First of all, we observe that COX brings more benefits when correspondence is more scarce.
        - This is because sufficient correspondence can maximally uncover the knowledge of OOM data. As correspondence gets less, the knowledge that can be explored from correspondence decreases.
        - However, COX leverages the knowledge from IM data which brings more benefits even with less correspondence.

For other detailed concerns, we have tried our best to fully address every single concern raised by each reviewer. Thanks to all reviewers for your valuable time in reviewing this paper, your help and support have largely improved this manuscript.

Since the interactive discussion phase has begun, we want you to know that **we are actively engaged in any discussion about any question, and will try our best to address all concerns or misunderstandings left.** Thank you so much for your time.

---

### Author Response · Authors · 2024-11-25
**We are here to address any remaining concerns**

Dear Reviewers, AC, and SAC:

We want to thank all reviewers for taking the time to review this paper, all of them have been greatly helpful and supportive. Moreover, we would like to thank AC and SAC for handling and guiding the reviewing process. Without your effort, we could not have reached so far.

The interactive discussion phase is one of the most effective features of ICLR, which allows us to clarify misunderstandings and improve the quality of papers. We highly value such an opportunity to be open-minded and learn from the experts in this field. So, before the discussion ends in two days, we hope if there are **any questions or suggestions** from **all** reviewers or AC, please don't hesitate to let us know. We will be truly grateful to hear your valuable opinions.

Sincerely,
Authors.

---

### Author Response · Authors · 2024-11-28
**Paper Revision**

Dear Reviewers and AC,

Thank you for all your constructive suggestions and comments. Here we have uploaded our revision on the following aspects:

- Incorporating all mentioned references and discussing their problem setting: **none of them** studies the problem of generalizing out-of-modality.
- Intuitively justifying our problem assumption through a **real-world** example, which is also consistent with the previous VIB **theoretical** framework.
- Providing detailed discussions of our training process based on the theoretical VIB framework.
- Incorporating additional baseline in the main comparison.
- Clarifying terms and notations.
- Conducting additional experiments to:
    - justify the benefits of leveraging IM data from IM perceptors.
    - study the effect of varied IM perceptor qualities on OOM performance.
    - testify the superiority of COX over competitive uni-modal baselines.

Moreover, for other specific questions or concerns, we have
- provided comprehensive details of our upper and lower bound derivation.
- release public source code to help understand our implementation.
- conducting additional ablations, such as various scales of IM perceptors, increasing modality numbers, and anchor methods.

We understand that you might be very busy at the time, if you could spare a little time to take a look at our revision, we would be genuinely grateful.

If any questions remain, we sincerely hope that you could kindly inform us. After all, we hope to make sure that no misunderstandings are left and a fair judgment of our contribution to this field can be made.

Sincerely,
Authors.

---

### Author Response · Authors · 2024-12-04
**Rebuttal Summary**

Dear Reviewers, AC, and SAC:

We deeply thank the hard work done by AC and SAC such as assigning reviewers, guiding the process, and further organizing the discussion. We also sincerely appreciate the reviewers for taking the time to read our paper, providing constructive opinions, and getting involved in our discussion. Without your elaborative help, our paper could not have been improved.

Here we summarize our rebuttal from a high-level perspective which could hopefully help grasp our contribution and modification quickly.

Through rebuttal, several consensuses have been achieved:

- Our paper is **interesting, practical, and meaningful**.
- Our explanation is **clear and helpful**, the intuition of this paper is **straightforward**.
- We have carefully addressed every question with proper justification such that 4 out of 5 reviewers are willing to further consider raising the score while the other reviewer stays positive.

To achieve this, we have strengthened our paper by:
- Justifying the effectiveness of COX through comparison with additional baseline and state-of-the-art uni-modal methods.
- Enhancing the related work discussion and details about the training process.
- Analyzing the benefits of both IM data and IM perceptors with additional studies.
- Clarifying the derivations and assumptions.

During the discussion phase, we tried our best to connect with reviewers and address every single concern, such as:
- Providing source code.
- Explaining the derivation details.
- Justifying our assumptions through both existing studies and real-world understandings.

It is such a great pleasure to know that **all reviewers** find our explanation helpful, and we will keep making our maximum effort to improve this paper by organizing the source code for open-source, polishing the assumptions with clearer demonstration, etc.

For any further internal discussion results, we kindly hope the AC could let us know in the meta-review, and we are more than happy to address them in our future revisions.

Thanks again for all your effort and contribution, we could not achieve this far without you.

Sincerely,
Authors.

---

### Meta-Review · Area_Chair_CDbX · 2024-12-17

**Metareview:**

This paper presents an early attempt to attack out-of-modal generalization issues in machine learning. It has experienced a fruitful author-reviewer discussion. All reviewers are now positive about this paper. I'm happy to suggest accepting this paper. Nonetheless, the authors are required to include the additional experiments and discussion in the final version.

**Additional Comments On Reviewer Discussion:**

All the reviewers are positive, with a score of 6 after rebuttal. The authors did hard work during rebuttal four out of five reviewers raised their score. I'm happy to suggest accepting this paper.

---

### Decision · Program_Chairs · 2025-01-22

Accept (Poster)